**Buoyant calving and ice-contact lake evolution at Pasterze Glacier (Austria) in the period**
**1998-2019**
Andreas Kellerer-Pirklbauer (1), Michael Avian (2), Douglas I. Benn (3), Felix Bernsteiner (1),
Philipp Krisch (1), Christian Ziesler (1)
(1) Cascade - The mountain processes and mountain hazards group, Institute of Geography and
Regional Science, University of Graz, Austria
(2) Department of Earth Observation, Zentralanstalt für Meteorologie und Geodynamik
(ZAMG), Vienna, Austria
(3) School of Geography and Geosciences, University of St Andrews, St Andrews, UK
**Correspondence**
Andreas Kellerer-Pirklbauer; andreas.kellerer@uni-graz.at
**Funding information**
Research relevant for this study was funded through different projects: (a)
Austrian Science Fund, project no. FWF P18304-N10, (b) Hohe Tauern National Park authority
(various projects), (c) Glockner Ökofonds (GROHAG) 2018, and (d) Austrian Alpine Association
(through the annual glacier monitoring program)
**Abstract:** Rapid growth of proglacial lakes in the current warming climate can pose significant
outburst flood hazards, increase rates of ice mass loss, and alter the dynamic state of glaciers.
We studied the nature and rate of proglacial lake evolution at Pasterze Glacier (Austria) in the
period 1998-2019 using different remote sensing (photogrammetry, laserscanning) and
fieldwork based (GNSS, time-lapse photography, geoelectrical resistivity tomography/ERT, and
bathymetry) data. Glacier thinning below the spillway level and glacier recession caused
flooding of the glacier, initially forming a glacier-lateral to supraglacial lake with subaerial and
subaquatic debris-covered dead-ice bodies. The observed lake size increase in 1998-2019
followed an exponential curve (1998: 1900 m²; 2019: 304,000 m²). ERT data from 2015 to 2019
revealed widespread existence of massive dead-ice bodies exceeding 25 m in thickness near the
lake shore. Several large-scale and rapidly occurring buoyant calving events were detected in
the 48 m deep basin by time-lapse photography, indicating that buoyant calving is a crucial
process for the fast lake expansion. Estimations of the ice volume losses by buoyant calving and
by subaerial ablation at a 0.35 km² large lake-proximal section of the glacier reveal comparable
values for both processes (c.1 x $10^6$ m³) for the period August 2018 to August 2019. We
identified a sequence of processes: glacier recession into a basin and glacier thinning below
spillway-level; glacio-fluvial sedimentation in the glacial-proglacial transition zone covering
dead ice; initial formation and accelerating enlargement of a glacier-lateral to supraglacial lake
by ablation of glacier ice and debris-covered dead ice forming thermokarst features; increase in
hydrostatic disequilibrium leading to destabilization of ice at the lake bottom or at the near-
shore causing fracturing, tilting, disintegration or emergence of new icebergs due to buoyant
calving; and gradual melting of icebergs along with iceberg capsizing events. We conclude that
buoyant calving, previously not reported from the European Alps, might play an important role
at alpine glaciers in the future as many glaciers are expected to recede into valley or cirque
overdeepenings.
**Keywords:** ice-contact lake; dead ice decay; buoyant calving; hydrostatic equilibrium; proglacial
landscape evolution
**1. INTRODUCTION**
Ongoing recession of mountain glaciers worldwide reveals dynamic landscapes exposed to high
rates of geomorphological and hydrological changes (Carrivick and Heckmann, 2017). In suitable
topographic conditions, proglacial lakes may form, including ice-contact lakes (physically
attached to an ice margin) and ice-marginal lakes (lakes detached from or immediately beyond
a contemporary ice margin) (Benn and Evans, 2010; Carrivick and Tweed, 2013). Such lakes
have increased in number, size and volume around the world due to climate warming-induced
glacier melt (Carrivick and Tweed, 2013; Otto, 2019). Buckel et al. (2018) for instance studied
the formation and distribution of proglacial lakes since the Little Ice Age (LIA) in Austria
revealing a continuous acceleration in the number of glacier-related lakes particularly since the
turn of the 21st century.

The formation of proglacial lakes is important because they can pose significant outburst flood
hazards (e.g. Richardson and Reynolds, 2000; Harrison et al., 2018), increase rates of ice mass
loss, and alter the dynamic state of glaciers (e.g. Kirkbride and Warren, 1999; King et al., 2018,
2019; Liu et al., 2020). However, detailed descriptions of proglacial lake formation and related
subaerial and subaquatic processes are still rare. Carrivick and Heckmann (2017) pointed out
that there is an urgent need for inventories of proglacial systems including lakes to form a
baseline from which changes could be detected.

The evolution of proglacial lakes is commonly linked to the subsurface, particularly to changes
in the distribution of debris-covered dead ice (defined here as any part of a glacier which has
ceased to flow) and permafrost-related ground ice bodies (Bosson et al., 2015; Gärtner-Roer
and Bast, 2019) affecting lake geometry and areal expansion.

Water bodies at the glacier surface form initially as supraglacial lakes which might be either
perched lakes (i.e. above the hydrological base level of the glacier) or base-level lakes (spillway
controlled). The former type is prone to drainage if the perched lake connects to the englacial
conduit system (Benn et al., 2001). Rapid areal expansion of such lakes is controlled by
waterline and subaerial melting of exposed ice cliffs and calving (Benn et al., 2001).
Furthermore, supraglacial lakes may transform into proglacial lakes lacking any ice core (full-
depth lakes) through melting of lake-bottom ice. However, this is a slow process in which
energy is conducted from the overlying water and cannot account for some observed instances
of fast lake-bottom lowering with rates exceeding 10 m yr$^{-1}$ (Thompson et al., 2012). It has been
argued that fast lake-bottom lowering could occur by buoyant calving (Dykes et al., 2010;
Thompson et al., 2012), but the rare and episodic nature of such events mean that little is
known about how buoyant calving might contribute to the transformation of supraglacial lakes
into full-depth lakes.

Buoyant calving occurs where ice is subject to net upward buoyant forces sufficient to
overcome its tensile strength. Such forces can develop where either ice thinning (e.g. via
surface ablation) or water deepening (e.g. rises in lake level) cause the ice to become buoyant.
If the ice is unable to adjust its geometry to achieve hydrostatic equilibrium it can become
super-buoyant (Benn et al., 2007), creating tensile stresses at the ice base. If these stresses
become sufficiently high, the ice will fracture and calve, as described by Holdsworth (1973),
Warren et al. (2001) and Boyce et al. (2007). Detailed models of super-buoyancy and buoyant
calving have been presented by Wagner et al. (2016) and Benn et al. (2017). Hydrostatic
disequilibrium caused the sudden disintegration of debris-covered dead ice in the proglacial
area of Pasterze Glacier in September 2016 (Fig. 2). This event was briefly described in Kellerer-
Pirklbauer et al. (2017) and was one of the main motivations for the present study.

In this study, we analysed rates and processes of glacier recession and formation and evolution
of an ice-contact lake at Pasterze Glacier, Austria, over a period of 22 years. The aims of this
study are (i) to examine glaciological and morphological changes at the highly dynamic glacial-
proglacial transition zone of the receding Pasterze Glacier and (ii) to discuss related processes
which formed the proglacial lake named Pasterzensee (*See* is German for lake) during the
period 1998-2019. Regarding the latter, we focus particularly on the significance of buoyant
calving. In doing so, we consider subaerial, subsurface, aquatic, as well as subaquatic domains
applying fieldwork-based and remote-sensing techniques.

**2. STUDY AREA**
The study area comprises the glacial-proglacial transition zone of Pasterze Glacier, Austria. This
glacier covered 26.5 km² during the LIA maximum around 1850 and is the largest glacier in the
Austrian Alps with an area of 15.4 km² in 2019 (Fig. 1). The glacier is located in the Glockner
Mountains, Hohe Tauern Range, at 47°05'N and 12°43'E (Fig. 1b). The gently sloping, 4.5 km
long glacier tongue is connected to the upper part of the glacier by an icefall named
Hufeisenbruch (meaning 'horseshoe icefall' in German) attributed to its former shape in plan
view. This icefall disintegrated and narrowed substantially during the last decades attributed to
the decrease of ice replenishment from the upper to the lower part of the glacier (Kellerer-
Pirklbauer, et al. 2008; Kaufmann et al., 2015).

The longest time series of length changes at Austrian glaciers has been compiled for Pasterze
Glacier. Measurements at this glacier were initiated in 1879 and interrupted in only three years.
Furthermore, annual glacier flow velocity measurements and surface elevation changes at
cross-sections were initiated in the 1920s with almost continuous measurements since then
(Lieb and Kellerer-Pirklbauer, 2018, chapter 4.2.). Technical details of the measurement can be
found in Kellerer-Pirklbauer et al. (2008) and Lieb and Kellerer-Pirklbauer (2018). Minor glacier
advances at Pasterze Glacier occurred in only seven years since 1879, the most recent of which
was in the 1930s. Even during wetter and cooler periods (1890s, 1920s and 1965-1980), the
glacier did not advance substantially, which can be attributed to the long response time of the
glacier (Zuo and Oerlemans, 1997). In 1959-2019, Pasterze Glacier receded by 1550 m, three
times the mean value for all Austrian glaciers (520 m), related to its large size. Today, Pasterze
Glacier is characterised by annual mean recession rates in the order of 40 m $yr^{-1}$ (Lieb and
Kellerer-Pirklbauer, 2018) causing a rather high pace of glacial to proglacial landscape
modification favouring paraglacial response processes (Ballantyne, 2002; Avian et al., 2018).

Analyses of brittle and ductile structures at the surface of the glacier tongue revealed that
many of these structures are relict and independent from current glacier motion (Kellerer-
Pirklbauer and Kulmer, 2019). The glacier tongue is in a state of rapid decay and thinning and
thus prone to fracturing by normal fault formation. Englacial and subglacial melting of glacier
ice caused the formation of circular collapse structures with concentric crevasses, which form
when the ice between the glacier surface and the roof of water channels decreases. Kellerer-
Pirklbauer and Kulmer (2019) concluded that the tongue of the Pasterze Glacier is currently
turning into a large dead-ice body characterized by a strong decrease in ice replenishment from
further up-glacier, movement cessation, accelerated thinning and ice disintegration by supra-,
en- and subglacial ablation, allowing normal fractures and circular collapse features to develop.
This rapid deglaciation and decrease in activity are favourable for dead ice and proglacial lake
formation.

An automatic weather station is located close to the study area operated by Austrian Hydro
Powers since 1982 (AWS in Fig. 1a). The coldest calendar year in the period 1998-2019 was
2005 with a mean annual air temperature (MAAT) of 0.9°C whereas the warmest year was 2015
with 4.0°C (range 3.1°C, mean of the 22-year period 2.4°C; Fig. 1c). Interannual variation is high
although a warming trend is clear. A MAAT value >3°C was calculated for eight of the nine years
between 2011 and 2019. No such high MAAT values were recorded for the entire previous 28-
year period 1982-2010 indicating significant recent atmospheric warming. Two ground
temperature monitoring sites were installed near the lake in fluvio-glacial sediments in 2018
(PRO1 – one sensor at the surface; PRO2 – three sensors at the surface and at 10 and 40 cm
depths; location see Fig. 1a) using GeoPrecision data logger equipped with PT1000 temperature
sensors (accuracy of +/−0.05°C) and logging hourly. Positive mean values for a 363-day long
period (13.09.2018-10.09.2019) were recorded for both sites (PRO1: 2.6°C, PRO2: 3.7-3.9°C)
suggesting permafrost-free conditions in the proglacial area and unfavourable conditions for
long-term dead ice conservation even below a protecting sediment cover.

**3. MATERIAL AND METHODS**
**3.1. GNSS data**
The terminus position of Pasterze Glacier was measured directly in the field by Global
Navigation Satellite System (GNSS) techniques in 14 years between 2003 and 2019 (annually
between 2003 and 2005, in 2008, and annually between 2010 and 2019). Direct measurements
of the subaerial glacier limit are essential in areas where debris cover obscures the glacier
margin, hindering the successful application of remote-sensing techniques (e.g. Kaufmann et
al., 2015; Avian et al., 2020). GNSS measurements were mostly carried out in September of the
above listed years, thus, close to the end of the glaciological years of mid-latitude mountain
regions. Until 2013, conventional GNSS technique was applied using different handheld
GARMIN devices (geometric accuracy in the range of meters). Afterwards, real time kinematics
(RTK) technique was used, where correction data from the base station whose location is
precisely known are transmitted to the rover (geometric accuracy in the range of centimetres).
We utilized a TOPCON HiPer V Differential GPS system. The base station was either our own
local station (base-and-rover setup) or we obtained correction signals from a national
correction-data provider (EPOSA, Vienna).

**3.2. Airborne photogrammetry and land cover classification**
Nine sets of high-resolution optical images with a geometric resolution of 0.09-0.5 m derived
from aerial surveys between 1998 and 2019 (Table 1) were available for land cover analyses.
For the years 2003, 2006, and 2009, the planimetric accuracy of single point measurements is
better than ±20 cm (Kaufmann et al., 2015). Comparable planimetric accuracies can be
expected for the other stages. The optical data sets were used for visual classification using a
hierarchical interpretation key following a scheme developed for Pasterze Glacier by Avian et al.
(2018) for laserscanning data and modified later for optical data by Krisch and Kellerer-
Pirklbauer (2019, Table 2 therein). Land cover classification was accomplished at a scale of
1:300 (for the stages 1998-2015; data based on Krisch and Kellerer-Pirklbauer, 2019) or 1:200
(2018-2019; this study). The classification results for a 1.77 km² area at Pasterze Glacier were
published earlier by Krisch and Kellerer-Pirklbauer (2019, Fig. 3 therein) for 1998, 2003, 2006,
2009, 2012, and 2015. For a 0.37 km² area, manual land cover classification was accomplished
in this study for 2018 and 2019 using the same mapping key.

**3.3. Terrestrial laserscanning**

The glacial-proglacial transition zone of Pasterze Glacier has been monitored by terrestrial
laserscanning (TLS) since 2001 from the scanning position Franz-Josefs-Höhe (FJH). The area of
interest in the scan sector covers 1.2 km² (Fig. 1a). Using scanning position FJH, one minor
limitation of TLS-based data for glacier lake delineation is the oblique scan geometry causing
data gaps due to scan-shadowed areas (Avian et al., 2018; 2020). Until 2009 the Riegl LPM-2k
system was used followed by the Riegl LMS-Z620 system since then. Technical specifications
regarding the two Riegl laserscanning systems as well as the configuration of the geodetic
network (scanning position and reference points) can be found in Avian et al. (2018). Processing
and registration of the TLS data (point clouds) was performed in Riegl RiScan, subsequently
DTMs (with 1 or 0.5 m grid resolution) were calculated in Golden Software Surfer. In this study
we used the DTMs to delineate the water bodies in the scan sector manually (for details see
Avian et al., 2020) supported by GNSS data (cf. above) for the glacier boundary. In addition, the
point clouds acquired by TLS were used to quantify lake level variations (see section 3.4). TLS-
data from 2010 to 2019 (13.09.2010, 27.09.2011, 07.09.2012, 24.08.2013, 09.09.2014,
12.09.2015, 27.08.2016, 22.09.2017, 13.09.2018, and 03.08.2019) were analysed.

Furthermore, we quantified ice-surface elevation changes of Pasterze Glacier near the
proglacial lake using TLS-data from 13.09.2018 and 03.08.2019. This was done to bring ice
volume losses by ablation at the lake-proximal part of the glacier in relation to ice mass losses
by buoyant calving for the period of (roughly) August 2018 to August 2019 (see below).
Although this data set does not cover an entire glaciological year, at least information about the
order of magnitude of the spatially distributed direct ice mass losses by subaerial ablation near
the shores of Lake Pasterzensee is gained. The emergence velocity as well as the general glacier
motion at the glacier terminus is close to zero (Kellerer-Pirklbauer et al., 2008; Kellerer-
Pirklbauer and Kulmer, 2019) apart from ice movement related to crevasses or steeper sloping
areas (Seier et al., 2017). Therefore, we can assume that surface elevation changes at the
glacier terminus between the two stages equals basically glacier ablation.

**3.4. Time-lapse photography**
At Pasterze Glacier six remote digital cameras (RDC) are installed to monitor mainly
glaciological processes with a very high temporal resolution (see Avian et al., 2020; overview
regarding the six cameras). One time-lapse camera was operated by the Grossglockner
Hochalpenstraße AG (GROHAG) using a Panomax system. The model used is a Roundshot
Livecam Generation 2 (Seitz, Switzerland) with a recording rate of mostly 5 minutes during
daylight. Time specification is UTC+2. The camera is installed at the Franz-Josefs-Höhe lookout
point (Fig. 1a) at an elevation of 2380 m asl and, thus, 310 m above the present lake level of
Lake Pasterzensee. Based on this optical data, Kellerer-Pirklbauer et al. (2017) reported a
sudden ice-disintegration event at the glacier lake in September 2016 where tilting, lateral
shifting, and subsidence of the ground accompanied by complete ice disintegration of a debris-
covered ice body occurred. For this study, we visually checked all available Panomax images
from 2016 to 2019. Four large-scale and rapidly occurring ice-breakup events (IBE) were
detected in the period September 2016 to October 2019 (IBE1: 20.09.2016; IBE2: 09.08.2018,
IBE3: 26.09.2018, IBE4: 24.10.2018). The effects on the proglacial landscape during these four
IBE was quantitatively analysed as follows.

For the orthorectification process of the Panomax images (7030x2048 px) it is necessary to find
a suitable mathematical model. To get the necessary parameters for this model, control points
are needed which are visible in both the Panomax images and pre-existing orthophotos used
for the orthorectification process. We applied an interpolation approach using the rubber
sheeting model in ERDAS IMAGINE 2018. This model calculates a Triangulated Irregular
Network (TIN) for all control points at the reference orthophoto and at the Panomax image and
transforms the calculated triangles of the oblique images in such a way that they equal the ones
of the reference orthophoto. First degree polynomials were used for the transformation within
the triangles. Only control points at the lake level were utilized to achieve a maximum accuracy
at lake-level objects. Reasons for minor geometric errors in the analysed orthorectified images
were changes in the lake level or an offset of the camera (maximum of 5 pixels).

To assess the potential effect of lake level changes on geometric errors in the orthorectified
images, we quantified lake-level variations by using GNSS and TLS data. We compared lake-level
data from nine different GNSS campaigns over a 5-year period (17.09.2015-22.09.2020; all from
the period between 11 am to 3 pm). Geometric accuracy is in the range of centimetres based
on comparison with stable points. Results yield a mean elevation of 2069.54 m asl ranging from
2069.87 asl (17.09.2015) to 2069.19 m asl (22.09.2020), thus a range of 0.68 m with a tendency
of lake-level lowering over time (Fig. 4c). In addition, we measured the elevation of small and
fresh-looking lake terraces next to the glacier terminus on 14.09.2020 with GNSS yielding an
elevation range of 0.59 m. This small elevation range is also in accordance with the lake-level
elevations measured by GNSS during two consecutive field campaigns on 14.09.2020 and
22.09.2020 with a difference of 0.53 m. TLS-based lake level estimation was accomplished for
six dates in the period 2014-2019 (see section 3.3.) by identifying the lowest level of the point
cloud at the lake shore (mean elevation of lower most measurement points at the lake shore).
Based on TLS data we observed a lake level variation in the order of 0.8 m and a trend in lake
level lowering during this period. Therefore, as judged from our long-term as well as short-term
GNSS and TLS data, we demonstrate rather stable lake-outflow as well as lake-level conditions
at least for the period 2015-2020 with a lake-level lowering trend. The assumption of long-term
lake level variations <1 m during the summer months (seasonal amplitude) is further supported
by field observations during the last years with the shape (stepped geometry) and size (< 1m
vertical extent) of thermo-erosional notches at the waterline. Therefore, the potential effect of
lake level changes on geometric errors in the orthorectified images should be small.

Three groups of control points were generated using the three pre-existing orthophotos of
11.07.2015, 11.09.2018, and 15.11.2018 (Table 1) and suitable Panomax images from the same
days. For the IBE1 we used the model of 11.07.2015, for IBE2 and IBE3 the model of
11.09.2018, and for IBE 4 the one of 15.11.2018. The calculated orthorectified images have a
geometric resolution of 0.2 m. ArcGIS 10.5 was subsequently used to analyse landform changes.

**3.5. Quantification of Quantification of ice mass losses by buoyant calving**
A quantification of ice losses by buoyant calving was attempted by using the Panomax images.
Three of the large-scale ice-breakup events occurred between August and September 2018
(IBE2 to IBE4). For these events we estimated the volume of the newly emerging icebergs and
the volume of uplifted ice masses detaching from the subaquatic glacier ice. The latter was
accomplished by comparing the calculated volume of a given ice-mass (e.g. a debris-covered ice
slab) before and after the ice-breakup event. For volumetric calculations we applied the
following approach. The horizontal extent of affected (newly emerged or uplifted) ice masses
was transferred back to and drawn into the original webcam images. A maximum iceberg
height was also drawn as a line in the original webcam image. The length of this line was then
quantified by using the ratio between the quantified horizontal extent and the marked line. The
iceberg height then was obtained by applying a correction calculation for the camera distortion
produced by an incidence angle of 25° (calculated by a height difference of 310m and a
horizontal distance of approx. 650m).

The volume of individual icebergs was approximated by assuming that all ice bodies above the
waterline have the form of a truncated pyramid, where A2 is 20% (for dome-shaped iceberg),
50% (for mixed iceberg type) or 80% (for tabular iceberg) of A1. The volume of truncated
pyramid (iceberg above the waterline) with irregular base is given by

$V = \frac{h}{3}\left(A_1 + \sqrt{A_1 * A_2} + A_2\right)$            (1)

with $A_1$ = area at the waterline (larger base), $A_2$ = area of the top face (smaller base; in our cases
20, 50 or 80% of $A_1$ depending on iceberg type), and h = maximum height of iceberg or
truncated pyramid (Harris and Stöcker, 1998). With this approach we quantified the volume of
nine icebergs for IBE2 (09.08.2018), eight for IBE3 (26.09.2018), and two for IBE4 (24.10.2018),
respectively. The volume above the waterline was then multiplied by 10 to calculate the total
iceberg volume. Significant uncertainties in this quantification attempt are the visual and thus
subjective estimation of the iceberg height and the fact that only large icebergs are considered.
Therefore, results of this approach must be seen only as order of magnitudes of ice mass losses
by buoyant calving in the period 09.08.2018 to 24.10.2018.

**3.6. Electrical resistivity tomography**
Electrical resistivity tomography (ERT) and seismic refraction (SR) has been applied in the study
area between 2015 and 2019. For space reasons, we focus only on selected aspects of the ERT
results in this paper. Electrical resistivity is a physical parameter related to the chemical
composition of a material and its porosity, temperature, water and ice content (Kneisel and
Hauck 2008). For ERT a multielectrode and multichannel system (GeoTom 2D system, Geolog,
Germany) and two-dimensional data inversion (Res2Dinv) using finite difference forward
modelling and quasi-Newton inversion techniques (Loke and Parker, 1996) was applied. ERT
was carried out at a total of 43 profiles (3 in 2015, 4 in 2016, 4 in 2017 [Fig. 3a,b], 5 in 2018, and
27 in 2019 [Fig. 3c]) with 2 or 4 m electrode spacing and profile lengths of 80-196 m. Salt water
was sometimes used at the electrodes to improve electrical contact. RTK-GNSS was applied to
measure the position of each electrode and thus the course of the profile (Fig. 3b). We applied
in most cases both the Wenner and Schlumberger arrays (Kneisel and Hauck, 2008). Focus is
given here on the Wenner results, which are more suitable for layered structures (Kneisel and
Hauck 2008). ERT data from 2015 and 2016 were taken from Hirschmann (2017) and Seier et al.
(2017). The apparent resistivity data were inverted in Res2Dinv using the robust inversion
modelling. ERT data were checked before processing for abnormally high or low resistivity
values. Abnormal values are commonly related to measurement errors and/or bad electrode
contact usually visible at all depths. Such 'bad datum points' were excluded manually (Kneisel
and Hauck, 2008). The number of iterations was stopped when the change in the RMS error
between two iterations was small.

**3.7. Bathymetry**
Sonar measurements were carried out at Lake Pasterzensee at the 13.09.2019. Water depth in
the lake was measured with a Deeper Smart Sonar CHIRP+ system (depth range 0.15-100 m)
consisting of an echo sounding device (single-beam echo sounder) and a GNSS positioning
sensor. CHIRP stands for Compressed High Intensity Radar Pulse. We measured with 290 kHz
(cone angle 16°) and a sonar scan rate of up to 15/second. According to the producer, the 16°
beam angle of the 290 kHz frequency results in a ground footprint of 0.28 m at 1m water depth,
of 2.81 m at 10 m water depth and of 11.24 m at 40 m water depth. These footprint values are
not optimal for resolving small-scale features at large water depths. However, as it was
intended in this study, the footprint values are acceptable for getting an overview of the lake
geometry.

The accuracy of raw water-depth measurements depends on the used device, beam angle,
sonar stability, bottom composition, and structure. Bandini et al. (2018) compared the Deeper
Smart Sensor PROx system (precursor of CHIRP+) against the ground truth. Their results
indicate a mean absolute error of 0.52 m for water depths of up to 30 m with almost perfect fit
(ground truth vs. sonar) at shallow sites. The tested PROx system underestimated the water
depth attributed to the beam diameter as it tends to take the shallowest point in the beam as
the depth reading when going over holes or slopes. No such comparative studies are published
for the CHIRP+ system. However, according to the producer the absolute error should be lower
for the CHIRP+ (pers. comm. by the technical support of Deeper, 16.12.2020). In conclusion, the
estimated accuracy of raw water-depth measurements should be less than 0.1 m at shallow (<5
m) and flat sites but might be as high as 0.5 m for deeper and sloping locations.

The CHIRP+ system was mounted on a Styrofoam platform for stability reasons and dragged
behind a small (and rather unstable) inflatable canoe operated by two people. Altogether 4276
water depth measurements along a 4.3 km long route were accomplished (Fig. 1d). Because
icebergs and wind cause boat instability, the canoe was not navigated along a regular shore-to-
shore route but rather in a zigzag mode starting in the northwest of the lake and ending in the
southeast. GNSS and water depth data were imported into ArcGIS for further analysis. To
compute the lake geometry, the measured lake depth values and a lake mask of September
2019 were combined using the Topo to Raster interpolation tool to calculate a digital terrain
model (DTM) with a 5m grid resolution. Lake volume was calculated using the functional
surface toolset.

**4. RESULTS**

**4.1. Glacier recession and areal expansion of the lake**



Figure 4a depicts the terminus positions between 1998 and 2019 as well as the proglacial water
surfaces including Lake Pasterzensee and the proglacial basin as defined for September 2019
(area of 0.365 km²). The glacier steadily receded into the current proglacial basin over a
longitudinal distance of about 1.4 km. In detail, however, this recession was not evenly
distributed along the glacier margin due to differential ablation below the uneven supraglacial
debris. The east part of the glacier tongue receded up-valley beyond the proglacial basin. The
west part of the glacier tongue is still in contact with the proglacial lake and changed
morphologically rather little during the last two decades. Figure 4a also depicts 100 m wide
strips where mean values for longitudinal and lateral backwasting were calculated. Results are
shown in Fig. 4b. The longitudinal backwasting rate was between 29.0 and 217.2 m yr$^{-1}$, 2 to 19
times larger than the lateral backwasting rate of 7.3 to 13.2 m yr$^{-1}$. High annual longitudinal
backwasting rates were measured in most years when the glacier was in the basin. Since 2017,
this rate drastically dropped, presumably due to the detachment of the glacier from the lake.

Figure 5 illustrates glacier recession and the evolution of proglacial water bodies for the period
1998-2019 in relation to the 0.365 km² proglacial basin as defined for September 2019. An
animation showing the general evolution of the proglacial lake between 2010 and 2020 is
published in the supplement. In 1998 only 0.5% of the basin was covered by water (Fig. 5a). Up
to 2006, water surfaces still covered less than 5% of the basin (Fig. 5c). By 2009, this value
increased to 11.2% (Fig. 5d) and was rather constant until two years later (Fig. 5f). By 2016,
more than 50% of the basin was covered by water (Fig. 5k) and in 2019 water surfaces in the
basin covered 83.2% (Fig. 5n). The increase in water surface areas in the basin since 1998
follows an exponential curve (Fig. 6a). However, in single years this areal increase follows a
distinct pattern with enlargement of water surfaces during summer and a decrease in autumn
due to lake level lowering as revealed by field observations. The exceptionally low value of
November 2018 (62.4%) in relation to September 2018 (73.2%) is related to the widespread
existence of ice floes. Figure 6a also depicts the extent of icebergs in the proglacial basin with
values below 1% in most cases. High percentage values were only mapped for 15.11.2018
(7.3%) followed by rapid iceberg loss during the ablation season 2019.

**4.2. Land cover change in the lake-proximal surrounding since 1998**
Different glacial and proglacial surface types and landforms were mapped for a 0.76 km² area in
the glacial-proglacial transition zone for nine different stages between 1998 and 2019 (Fig. 7).
The visual landform classification gives a more detailed picture on landform changes in the area
of interest. Figure 6b quantitatively summarises the relative changes of different surface types
in this transition zone. Debris-poor, rather clean-ice covered 58% of the area in 1998, decreased
to 9.3% until 2015, and vanished afterwards from the area. In contrast, debris-rich glacier parts
covered in all nine stages between 20.5% (2019) and 33.4% (2015) of the transition zone. For
this class, areal losses due to glacier recession were partly compensated by areal gains due to
an increase in supraglacial debris-covered areas. Water surfaces increased from 2.1% in 1998 to
45.5% in 2019. The low value for 15.11.2018 is related to ice floes (3.4%), data gaps (4.1%), as
well as high values for both debris-rich (2.1%) and debris-poor (1.5%) icebergs. Areas covered
by bedrock and vegetation were always around 4%. Areas covered by fine-grained sediments
reached a maximum in 2012 decreasing substantially afterwards (mainly due to lake extension).
Areas covered by coarse-grained sediments increased from 3.3% in 1998 to about 26-27% in
2018 and 2019 and are located at the northern and eastern margin of the basin. Finally, dead
ice holes were mapped for all stages, but their spatial extent was always very small (maximum
in 2012 with a total area of 618 m²) and covered less than 0.1% of the basin.

**4.3. Buoyant calving at the ice-contact lake**
Four large-scale ice-breakup events (IBE) related to buoyancy were detected for the period
September 2016 to October 2019 (IBE1: 20.09.2016; IBE2: 09.08.2018, IBE3: 26.09.2018, and
IBE4: 24.10.2018). Twelve smaller to mid-sized iceberg-tilting or capsize events were
additionally documented by the Panomax images (27.05.2017, 28.05.2017, 09.06.2017,

429  11.06.2017, 20.06.2017, 05.07.2017, 19.07.2017, 25.09.2017, 22.06.2018, 23.09.2018,

26.09.2018, and 30.10.2018).

IBE1 occurred on 20.09.2016. Figure 8a presents two ortho-images from this event at its
beginning (9:00 am) and its end (11:15 am). The latter also indicates the position of the
geoelectric profile ERT17-1 for orientation. Figure 2 visualizes the same event. An animation
depicting this ice-breakup event is published in the supplement. Different processes occurred
as indicated by the capital letters in Fig. 7a: Limnic transgression (A and F) of water due to
tilting of ice slabs, uplift of a debris-covered ice slab (B and G), formation of a massive crevasse
(C), complete ice disintegration (D), ice disintegration and lateral displacement of several ice
slabs (E), and drying out of a meltwater channel (H). All processes apart from the limnic
transgressions ended by 11:15 am, the latter terminated at 3:30 pm. The formation of the large
crevasse started initially at 9:30 am, followed by a rapid widening until 9:45 am (crack width 3.5
m), steady conditions until 10:45 am, followed by a second widening phase (crack width 5.5 m)
until 10:50 am (see inset graph in Fig. 8a). The morphologically most distinct event happened
between 9:50 am (Fig. 2d) and 9:55 am (Fig. 2e) when the total collapse of a 1700 m² large ice
slab occurred accompanied by lateral shift and tilting of neighbouring ice slabs by lateral push
(E) and lowering of the surface of previously tilted slabs (B).

IBE2 happened on 09.08.2018. Figure 8b depicts the changes that occurred between 4:35 pm
and 4:58 pm. At this event three different processes were identified: (A) detachment of a
debris-covered ice peninsula (945 m²) from Pasterze Glacier at the western lakeshore and
separation into four icebergs (total area 1054 m²), (B) emergence of a 1035 m² large iceberg
(4:35-4:40 pm) followed by capsizing and partially disintegration of this iceberg into ice debris
(4:40-4:58 pm) pushing away other icebergs which cause (C) lateral iceberg displacement of up
to 65.6 m as well as a clockwise iceberg rotation of 95°.

IBE3 occurred on 26.09.2018. This event involved four main processes as visualised in Fig. 8c:
(A) uplift of debris-covered ice bodies increasing the surface area from 6820 to 13245 m² in
only 10 minutes (at 2:35-2:45 pm), (B) emergence of a new iceberg between 2:35 and 2:40 pm
which capsized a few minutes afterwards, (C) limnic transgression, and (D) lateral iceberg
displacement (both at 2:35-3:00 pm). At the southern part of the affected area, icebergs moved
away from the uplifting area (push effect). In contrast, at the eastern part of the affected area
icebergs moved towards the uplifting area possibly due to compensatory currents causing a
suction effect. A large iceberg (IB1 in Fig. 7c) was hardly moving at all suggesting grounding
conditions.

The last major IBE took place on 24.10.2018 (IBE4) spanning only 5 minutes (Fig. 8d). Like IBE2,
a debris-covered ice peninsula (1,933 m²) detached from Pasterze Glacier at the western
lakeshore and separated into several icebergs (A). Furthermore, (B) ice disintegration and (C)
lateral iceberg displacement was observed during the event. The large iceberg IB1 experienced
a lateral offset of 22 m accompanied by a clockwise rotation by 43°. Spatial extent, volume and
freeboard of this iceberg were calculated based on a high-resolution DTM derived from the
aerial survey dating to 15.11.2018 (cf. Table 1). The subaerial volume of iceberg IB1 was 3271
m³ on 15.11.2018, which should be around 10% of the entire iceberg. Hence, some 29,500 m³
(90%) were during that time below the lake level. Maximum freeboard of IB1 was 3.7 m with a
mean freeboard value of 1.4 m. If we assume the same surface area of the iceberg below lake
level (2287 m²), we could further assume a mean ice thickness of the iceberg of 14.3 m (12.9 m
draft, 1.4 m freeboard). Therefore, in order to have a freely moveable iceberg, a water depth
exceeding 13 m is needed.

No large buoyant calving events were detectable in the time-lapse images after 24.10.2018.
However, at least the occurrence of small-sized buoyant calving events which are hardly
detectable by the time-lapse camera can be assumed. During field work in June 2019, we
observed buoyant calving of a small, c.3 m long iceberg ('shooter' according to Benn and Evans,
2010) c.200 m from the subaerial glacier front (Fig. 3d). The whole event took only few minutes
and was hardly visible in the time-lapse images of that particular day.

**4.4. Ice mass loss by buoyant calving and subaerial ablation**
The quantification of the ice loss by buoyant calving for the three events IB2 to IB4
approximated by ice detachment, uplift and emergence processes revealed the following
results. The sum of movement-affected ice masses (without lateral displacement) during the
three ice-breakup events were 55,717 m³ for IBE2, 445,257 m³ for IBE3, and 537,604 m³ for IBE,
respectively, summing up to 1,038,578 m³ (Table 2). As no other substantial ice break-up events
occurred afterwards, we can therefore assume that ice loss by buoyant calving in the period
August 2018 to August 2019 at Pasterze Glacier was at least in the order of $1 \times 10^6$ m³.

The comparison of the two sets of TLS-data from 13.09.2018 and 03.08.2019 revealed surface
elevation changes and thus more or less glacier ice ablation of up to 5 m between the two
stages. It was not the scope of this paper to analyse ablation rates at the terminus of Pasterze
Glacier in detail. However, for a rough estimate we can calculate for the lowest part of the
glacier tongue next to the proglacial lake (see Fig. 1, c.0.35 km²) the total ice loss for the period
September 2018 to August 2019. Mean ablation rates of 2.5 m or 3.0 m for this area would
yield total ice losses by ablation for this area of 870,000 m³ and 1,050,000 m³, respectively.

**4.5. Ground ice conditions at the lake basin and its proximity**

Altogether 43 ERT profiles were measured in the proglacial area between 2015 and 2019 with profile lengths of between 80 and 196 m. In this study we focus on the quantification of sediment-buried dead ice bodies detected by ERT. A detailed discussion on the ERT results will be presented elsewhere. Resistivity values >20,000 Ohm m indicate buried glacier ice and water-saturated glacial sediments show values <3,000 Ohm m (Pant and Reynolds, 2000). Clay and sand have resistivity values in the ranges of 1-100 and 100-5,000 Ohm m, respectively. Temperate glacier ice may exceed 1 x 10$^6$ Ohm m (Kneisel and Hauck, 2008). We used the 20,000 Ohm m-boundary in the interpretation to estimate the maximum ice thickness for each profile as depicted in Fig. 9 which shows three profiles from 2017. In many cases, ice thickness exceeded the depth of ERT penetration. Therefore, we only were able to calculate 'minimum ice thickness estimates' based on the ERT data.

Figure 10 summarises the results of the surveys for 2015, 2016, 2017, 2018 and September 2019. Two of the three ERT profiles measured in 2015 (ERT15-1, ERT15-2) revealed only very thin ice lenses. Both are located outside the proglacial basin as defined in September 2019 (Fig. 10a). The profile in the basin had an estimated ice thickness of 14 m (ERT15-3). The profiles measured in 2016 revealed minimum ice thickness values of 8-10 m (Fig. 10b). The four profiles measured in 2017 in the central part of the proglacial area revealed minimum ice thicknesses of between 13 (ERT17-4) and 28 m (ERT17-2) (Fig. 10c) confirming the existence of massive dead ice beneath a thin veneer of debris (Fig. 9).

The interpretation of four profiles measured in 2018 are shown in Fig. 10d. Profiles ERT18-2 and
ERT18-3 are free of ice located outside the basin or at its margin. ERT18-4 and ERT18-5 were
both located in the basin and revealed minimum ice thicknesses of 13 (ERT18-5) and 14 m
(ERT18-4). The September-2019 measurements supported earlier measurements (Fig. 10e). The
profiles at the eastern margin of the basin showed again a thin layer (ERT19-18; 8m ice) or only
very small occurrences of glacier ice (ERT19-19; 1 m ice). The three profiles near the north-
western shore of the lake revealed minimum ice thickness estimates of up to 26 m (ERT19-26).
In summary, ERT profiles outside the proglacial basin typically showed little buried dead ice
remnants, whereas profiles in the basin (particularly at its north-western part) typically yielded
resistivity values consistent with widespread massive dead ice.

**4.6. Bathymetry of the lake basin**
Lake bottom geometry and water volume of Lake Pasterzensee was calculated based on 4276
sonar measurements (Fig. 1d). Measured water depths ranged from 0.35 m to 48.2 m yielding
an arithmetic mean of 13.4 m and a median of 10.7 m. During the time of bathymetric
measurements, the lake level was 2069.1 m asl implying that the lowest point at the lake
bottom was 2020.9 m asl (Fig. 11a). Several sub-basins (marked as A-D in Fig. 11a) were
identified along the 1.2 km long and up to 300 m wide lake basin. One small sub-basin (A) was
detected close to the southern end of the lake with maximum measured water depths
exceeding 20 m (maximum 24.1 m, 2045 m asl), an E-W extent of 160 m, and a N-S dimension
of 140 m. A second sub-basin (B) is slightly less deep (max. 20.5 m) but seems to be broader
compared to basin (A). The third sub-basin (C) is by far the deepest, the largest, and the most
complex one with a maximum water depth of 48.2 m and a secondary basin in the south
reaching a measured maximum depth of 31.0 m. In this sub-basin, water depths exceeding 30
m were calculated for a 34,000 m² large in the central part of the entire lake basin. The lake
basin gets generally shallower towards the northwest. Finally, a fourth sub-basin (D) was
identified at the north-western end of Lake Pasterzensee where a broad basin is located with a
maximum measured depth of 17.7 m. Based on our gridded DTM for the lake bottom, the
estimated water volume of the 299,496 m² large Lake Pasterzensee in September 2019 was 4 x
$10^6$ m³. The gradient from the deep basin (C) to the shore seems to be rather gradual at the
eastern margin of the lake. In contrast, at the western margin of the lake basin where Lake
Pasterzensee is in ice-contact, the gradient is steep in most areas (e.g. at sub-basin C: horizontal
distance between sonar measurement location and glacier margin 19 m vs. water depth 26.1m)
suggesting a steep glacier margin with a pronounced ice foot.

**5. DISCUSSION**
**5.1. Glacial-to-proglacial landscape modification**
Pasterze Glacier receded by some 1.4 km between 1998 and 2019 thereby causing the
formation of a bedrock-dammed lake in an over-deepened glacial basin. During these two
decades, the glacier decelerated, fractured (Kellerer-Pirklbauer and Kulmer, 2019) and lost the
connection to the lake at its eastern part. In contrast, at the western shore, the lake was still in
ice contact with the glacier in 2019. This ice-contact difference is related to an unequal
recession pattern of the eastern and western part of the glacier tongue caused by an uneven
distribution of the supraglacial debris cover (Kellerer-Pirklbauer, 2008). The debris cover
distribution pattern promotes differential ablation (Kellerer-Pirklbauer et al., 2008). Rapid
deglaciation as well as glacier thinning is much more intensive at the debris-poor part of the
glacier affecting the stress and strain field and modifying the flow directions of the ice mass
(Kaufmann et al., 2015). Therefore, the proglacial lake predominantly developed in areas where
debris-poor ice was located before.

At the waterline, thermo-erosional undercutting causes the formation of notches (cf. Röhl,
2006). Such notches are frequent features at Pasterze Glacier and were first reported in 2004
(Kellerer-Pirklbauer, 2008). DPGS measurements at the glacier margin on 13.09.2019 showed
that waterline notches occurred during that time at 53% of the 935 m long ice-contact line
between Pasterze Glacier and Lake Pasterzensee (Fig. 5n). Notches observed at Pasterze Glacier
during several September-field-campaigns during the last years had a stepped geometry due to
lake-level drop. The amplitude of water-level fluctuations at Pasterzesee in the period 2015 to
2020 was less than a meter based on GNSS and TLS data indicating rather stable lake-outflow
conditions. However, GNSS and TLS data both show a lake-level lowering trend since 2015.

Stepped geometries were observed also at other alpine lakes (e.g. Röhl 2006). Rates of notch
formation and, thus, thermo-erosional undercutting at Pasterze Glacier are unknown. However,
if we consider the annual lateral backwasting rates derived from GNSS data (Fig. 4) as indicative
for thermo-erosional undercutting, a mean melt rate of about 10 m $yr^{-1}$ for the period 2010-
2019 can be assumed. This is about one third of the values quantified for Tasman Glacier (Röhl,
2006). The difference is possibly related to cooler (higher elevation) and more shaded (NE-
facing) conditions at Pasterze Glacier. Outward toppling of undercut ice masses due to thermal
erosion, a process potentially relevant for calving at ice-contact lakes (Benn and Evans 2010),
was not observed at Pasterze Glacier. Lateral backwasting at Pasterze Glacier is mainly
controlled by ice melting either beneath supraglacial debris or at bare ice cliffs above notches
where the slope is too steep to sustain a debris cover and thus the rock material slides into the
lake (see Fig. 10 in Kellerer-Pirklbauer, 2008).

The analysis of the relationship between glacier recession and the evolution of proglacial water
surfaces showed drastic changes in 1998-2019. The spatial extent of water surfaces in the 0.37
km² proglacial basin followed an exponential curve with 0.5% water surfaces in 1998, 21% by
2013, 51% by 2016, and 83% by 2019. On an annual timescale water surface changes follow a
distinct pattern with enlargement during summer due to glacier and dead-ice ablation in lake-
contact locations causing lake transgression and a shrinkage in size in autumn due to lake level
lowering. This annual pattern at Lake Pasterzensee has been also detected and quantified by
Sentinel-1 and Sentinel-2 data (Avian et al., 2020).

Carrivick and Tweed (2013) discuss the enhanced ablation at ice-contact lakes via mechanical
and thermal stresses at the glacier-water interfaces. They report increasing lake sizes in the
proglacial area of Tasersuaq Glacier, west Greenland, for four different stages between 1992
and 2010. An exponential increase in lake size, as observed at Pasterze Glacier, was however
not observed at Tasersuaq Glacier as judged from their provided map in the paper. More
general, detailed studies of increasing lake size on an annual basis are rare impeding the
comparison of our results with other studies accomplished in similar topoclimatical settings.
Some comparative observations are, however, as follows.

Schomacker and Kjær (2008) report from a glacier in Svalbard that an ice-contact lake increased
near-exponentially in size during a period of 40 years due to dead-ice melting. Schomacker
(2010) report from the enlargement of proglacial lakes at Vatnajökull in Iceland where the lake
Jökulsárlón enlarged by 40% in only 9 years (2000-2009). For the same lake, Canas et al. (2015)
revealed an enlargement by 74% for the period 1999-2014. Stokes et al. (2007) report an 57%
increase in the surface area of supra- and proglacial lakes in the Caucasus Mountains in the
period 1985-2000. Loriaux and Casassa (2012) described the evolution of glacial lakes from the
Northern Patagonia Icefield reporting a total lake area increase of 64.9% in a 66-year period
(1945-2011). Gardelle et al. (2011) detected for the Eastern Himalaya an enlargement of glacial
lakes by 20% to 65% between 1990 and 2009. To conclude, the numbers summarised here
clearly show that the increase in lake size at Pasterze Glacier is particularly high although this
relative increase in area at Lake Pasterzensee is likely biased by the very small initial size of the
lake in 1998.

Landscape changes were quantified for a 0.76 km² large transition zone between Pasterze
Glacier and its foreland for the period 1998-2019. Apart from rapid deglaciation and lake size
increase, areas covered by coarse-grained glacio-fluvial sediments increased in their extent.
Furthermore, icebergs in the lake were mapped for the first time in 2015 (0.7% of the 0.76 km²
large area) and reached their maximum extent in 2018 (3.5%). By the end of the ablation
season in 2019, the areal extent of icebergs decreased dramatically to only 0.3% attributed to
high melt rates in a warm summer 2019 (Fig. 1c: the MAAT in 2019 was the second highest in
the period 1998-2019). After 2015, an alluvial fan with a lake delta developed at the northern
end of the lake because the glacier receded at this location from the lake basin connecting the
main glacial stream directly with the lake (Fig. 6f and g). This recession was, however, only
superficial, and huge amounts of dead ice remained in the basin – as detected by ERT
measurements – and were covered by fluvio-glacial sediments.

**5.2. Dead-ice conditions and changes**
Subsurface conditions at the proglacial area of Pasterze Glacier were studied by measuring
electrical resistivity along 43 profiles distributed over the entire proglacial area between 2015
and 2019. Our measurements showed that dead ice bodies covered by sediments were absent
outside the proglacial basin as defined for September 2019. In contrast, all ERT measurements
carried out in the basin revealed very high maximum and median resistivity values (e.g. Fig. 9)
indicative of buried ice. Long-term air temperature data from a nearby automatic weather
station as well as two ground temperature data series directly from the proglacial area clearly
suggest that permafrost is absent at the shores of Lake Pasterzensee due to permafrost-
unfavourable thermal conditions (MAAT always >2.5°C since 2011). Furthermore, a distinct
warming trend occurred in the period 1998-2019 at Pasterze Glacier enhancing ice ablation and
deglaciation processes at the surface and the surface in more recent years.

In addition to the geomorphic observations made at the surface such as dead-ice holes (Figs 6b
and 7) or cracks (Fig. 2) in hummocky fluvio-glacial sediments (Fig. 3c), our subsurface data
clearly suggest substantial and rapid dead-ice degradation at present. Gärtner-Roer and Bast
(2019) conclude that only a few attempts have been made to describe and analyse the
occurrence, distribution, and dynamics of ground ice in recently deglaciated areas. However,
due to the rapid increase in proglacial areas at present, these authors point out that there is
increasing interest on research both for geomorphologist and hydrologists. With the presented
geophysical data from Pasterze Glacier, we proved the widespread existence of debris-covered
dead-ice bodies in a proglacial basin of an alpine valley glacier and, thus, contribute to this
emerging topic.

**5.3. Ice-breakup and buoyant calving**
Four remarkable ice-breakup events (IBE) with horizontal extents in the order of hundreds of
meters occurred in the period September 2016 to October 2018. No comparable events were
observed before the 20.09.2016 (Kellerer-Pirklbauer et al., 2017) and no comparable event
happened between 25.10.2018 and 29.07.2020. Only smaller buoyant calving events can be
assumed for the latter period as suggested by a fortuitously observed event (Fig. 3d).
Approximations of the ice volume lost by buoyant calving as well as by ablation through
subaerial melting at the lowest part of Pasterze Glacier have been in the same order of
magnitude (c.1 x $10^6$ m³) in almost identical periods (for buoyant calving: August 2018 to August
2019; for subaerial melting: September 2018 to August 2019). However, as the period August to
October 2018 was very unusual in terms of larger ice-breakup events (three of the four large
events occurred in this period), we can clearly conclude that multiannual glacier ice losses by
buoyant calving are substantial smaller compared to subaerial ablation rates.

Our field observations show that sediment is present on top of dead ice, particularly at the
north-western end of the lake where the main glacial stream enters the lake. Sediment cover
will affect the buoyant weight of the ice column, potentially offsetting buoyant forces and
inhibiting calving. It is not possible to quantify this effect, due to limited data on sediment and
ice thicknesses. It is clear, however, that although sediment cover will have delayed the onset
of buoyant calving, it was insufficient to prevent it in this case.

Thanks to high-resolution (both spatial and temporal) time-lapse photography overlooking the
glacial-proglacial transition zone, different ice-related processes can be clearly distinguished.
Common features of the IBEs are (a) limnic transgression due to ice slab lowering or tilting, (b)
drying out of meltwater channels due to slab uplift or tilting of ice slabs, (c) uplift – and
therefore enlargement – of previously existing ice-cored terraces or icebergs, (d) crack and
crevasse formation at previously stable-looking terraces, (e) sudden disintegration of ice
masses (i.e. collapsing ice masses) within minutes into ice debris, (f) lateral displacement of
icebergs (either pushed away or dragged towards uplifting icebergs), (g) emerging new icebergs
previously not mapped due to buoyant calving, (h) capsizing of new icebergs, and (i)
detachment of 'ice peninsulas' attached to Pasterze Glacier at the western lakeshore and
subsequent fragmentation into several icebergs and disintegration into small, mainly floating
icebergs. Regarding emergence of new icebergs, our observations suggest both buoyant calving
of small ice masses (suggested by emerging small icebergs, e.g. Fig. 3d) but also full-thickness
ice calving (suggested by the large ice-breakup events; Fig. 8).

All these processes are related to hydrostatic disequilibrium of the glacier margin or subaquatic
dead ice which becomes super-buoyant and subject to net upward buoyant forces (Benn et al.,
2007). Buoyant glacier margins can slowly move back into equilibrium by ice creep or can
fracture catastrophically as described for instance for Glacier Nef in Chile by Warren et al.
(2001). At Pasterze, creep rates are very low at the glacier margin with only few meters per
year near the terminus (Kellerer-Pirklbauer and Kulmer, 2019) therefore only the latter option
for a renewed hydrostatic equilibrium is feasible. A floating process of the glacier terminus was,
however, not observed at Pasterze Glacier (Boyce et al., 2007). Our buoyant calving
observations as well as the bathymetric data suggest the existence of an ice foot at the west
shore of the ice-contact lake. Such a presence of an ice foot below the water level of tidewater
ice cliffs of temperate glaciers has been debated for more than 120 years (Hunter and Powell,
1998). At Pasterze Glacier only small ice cliffs above thermo-erosional notches exist. However,
the existence of an ice foot at the western shore is very likely. This assumption is supported by
the occurrence of the ice breaking events with buoyant calving-related processes.

In summary, we identified the following sequence of processes at Pasterze Glacier: (a) glacier
recession into an overdeepened basin and glacier thinning below spillway-level; (b) glacio-
fluvial sedimentation in the glacial-proglacial transition zone covering dead ice; (c) initial
formation and accelerating enlargement of a glacier-lateral to supraglacial lake by ablation of
glacier ice and debris-covered dead ice forming thermokarst features; (d) increase in
hydrostatic disequilibrium leading to general glacier-ice instability; (e) destabilization of debris-
buried ice at the lake shore expressed by fracturing, tilting, and disintegration due to buoyancy;
(f) emergence of new icebergs due to buoyant calving; (g) gradual melting of icebergs along
with iceberg capsizing events. This sequence of processes is visualized in a conceptual model
depicted in Fig. 12. Our observations suggest that buoyant calving, previously not reported
from the European Alps, might play an important role at alpine glaciers in the future as many
glaciers are expected to recede into valley overdeepenings or cirques.

**6. CONCLUSIONS**
We studied the glacial-to-proglacial landscape transformation at the largest glacier in Austria
during the period 1998 to 2019 focusing on ice-contact lake evolution and buoyant calving
processes in an overdeepened basin. The main conclusions which can be drawn from this study
are the following:
• High annual backwasting rates were measured in most years when the glacier
terminated in the basin. The detachment of the glacier from the lake at the east side
drastically reduced backwasting rates.
• Detailed studies of increasing lake size on an annual basis are rare. We showed that the
increase in water surfaces in the basin since 1998 follows an exponential curve (1998:
1900 m²; 2019: 0.3 km²). The increase in lake size is particularly high although this
pattern is likely biased by the very small initial size of the lake in 1998. In single years
this areal increase follows a distinct pattern with enlargement of water surfaces during
summer and a decrease in autumn due to lake-level lowering supporting earlier
satellite-based studies (Avian et al. 2020).
• Icebergs in the up to 48.2 m deep lake were observed for the first time in 2015 and
reached their maximum extent in 2018. By the end of the ablation season in 2019, the
areal extent of icebergs decreased dramatically, attributed to high melt rates in a warm
summer 2019.
• Both, geomorphic observations made at the surface and geophysical data from the
subsurface clearly suggest widespread existence of debris-covered dead-ice bodies in
the proglacial basin which is substantially and rapidly affected by dead-ice degradation
at present due to permafrost-unfavourable ground temperature conditions.
• Previously, little was known about how buoyant calving might contribute to the
transformation of supraglacial lakes into full-depth lakes lacking any ice at the lake
bottom. Thanks to time-lapse images and photogrammetric data analysis, we were able
to analyse four large-scale ice-breakup events related to ice buoyancy for the period
September 2016 to October 2018. However, no large buoyant calving events were
detectable in the time-lapse images after 24.10.2018 and until (at least) 30.11.2020.
• Ice volumes lost by buoyant calving and by ablation through subaerial melting at the
lowest part of Pasterze Glacier revealed only for the period of (roughly) August 2018 to
August 2019 comparable values (c.1 x $10^6$ m³). In all other years, ice loss by buoyant
calving was substantially less important compared to subaerial ablation in terms of
volumetric effect. Although buoyant calving is not the most important ablation term in
the long term, it can result in large losses of ice and rapid geometric changes in the
short term. Buoyant calving can bring about a rapid transition of a lake from supraglacial

to full-depth and in some settings might cause a switch in the ablation regime, from

subaerial melt-dominated to full-depth calving dominated.

•   Different ice-related processes related to hydrostatic disequilibrium have been

identified: limnic transgression due to ice slab lowering or tilting; drying out of

meltwater channels due to slab uplift or tilting of ice slabs; uplift and enlargement of

ice-cored terraces or icebergs; crack formation at previously stable-looking terraces;

sudden disintegration of ice masses into ice debris; lateral displacement or rotation of

icebergs; emergence of new icebergs due to buoyant calving; capsizing of icebergs;

detachment of ice peninsulas attached to the glacier and subsequent fragmentation into

several icebergs.

•   Our observations suggest that buoyant calving, previously not reported from the

European Alps, might play an important role at alpine glaciers in the future as many

valley and cirque glaciers are expected to recede into valley overdeepenings or corries.


**Data availability.** Terminus position of Pasterze Glacier for the period 1998 to 2019, extent of
proglacial water surfaces between 1998 and 2019, and lake depth data from 13.09.2019 are
available in the Supplement.

**Supplement.** The supplement consists of three data sets and two animations: data sets: (1)
terminus position of Pasterze Glacier for the period 1998 to 2019, (2) extent of proglacial water
surfaces between 1998 and 2019, and (3) lake depth data based on echo sounding acquired on
13.09.2019; animations: (1) general evolution of the proglacial lake between 2010 and 2020
based on webcam images, and (2) ice-breakup event which occurred on the 20.09.2016. The
supplement related to this article is available online at: https://doi.org/10.5194/tc-2020-227-
supplement.

**Author contributions.** The study was designed by AKP. Fieldwork and analysis were carried out
by AKP (GNSS, geophysics, bathymetry), MA (laserscanning), FB (time-lapse photography), PK
(land cover mapping), CZ (geophysics, bathymetry). DIB contributed to the introduction and
discussion. AKP prepared the manuscript with contributions from all co-authors

**Competing interests.** The authors declare that they have no conflict of interest.

**Acknowledgments.** This study was funded by different projects over the years. The most
important ones are: (a) Austrian Science Fund, project no. FWF P18304-N10, (b) Hohe Tauern
National Park authority (several projects), (c) Glockner Ökofonds (GROHAG) 2018, and (d)
Austrian Alpine Association (through the annual glacier monitoring program). Meteorological
data were kindly provided by Austrian Hydro Powers. Aerial surveys of 2018 and 2019
(AeroMap) were funded by project (c) and the Institute of Geography and Regional Science
(supported by Wolfgang Sulzer). Matthias Wecht, Gernot Seier and Wolfgang Sulzer are very
much thanked for supporting the aerial photograph analysis of the two AeroMap flight
campaigns in 2018 and 2019. Correction signals for real time kinematics measurements were
kindly provided free of charge by EPOSA, Vienna. Field work was supported during numerous
field trips by several colleagues and numerous students especially Michael Bliem, Stefan
Brauchart, Alexander Dorić, Iris Hansche, Matthias Lichtenegger, Christian Lieb, Gerhard Karl
Lieb, Matthias Rathofer, Rupert Schwarzl, and Daniel Winkler. Melina Frießenbichler is kindly
acknowledged for processing TLS-data. Finally, the authors acknowledge the financial support
by the University of Graz.

**ORCID**
Andreas Kellerer-Pirklbauer https://orcid.org/0000-0002-2745-3953

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

**Tables and table captions**

**Table 1:** Technical parameters of aerial surveys between 1998 and 2019 used in this study. For
2003, 2006, and 2009 see also Kaufmann et al. (2015). KAGIS = GIS Service of the Regional
Government of Carinthia; BEV = Federal Office of Metrology and Surveying.

| Aerial survey | Acquisition date | Source | Geometric resolution of calculated orthophotos |
|---|---|---|---|
| 1998 | Aug. 1998 | National Park Hohe Tauen | 0.5 m |
| 2003 | 13.08.2003 | Kaufmann et al. (2015) | 0.5 m |
| 2006 | 22.09.2006 | Kaufmann et al. (2015) | 0.5 m |
| 2009 | 24.08.2009 | Kaufmann et al. (2015) | 0.5 m |
| 2012 | 18.08.2012 | KAGIS / BEV | 0.2 m |
| 2015 | 11.07.2015 | KAGIS / BEV | 0.2 m |
| 2018 | 11.09.2018 | KAGIS / BEV | 0.2 m |
| 2018 | 15.11.2018 | AeroMap GmbH | 0.1 m |
| 2019 | 21.09.2019 | AeroMap GmbH | 0.09 m |





**Table 2:** Affected ice masses during the three ice-breakup events IB2 (09.08.2018), IB3
(26.09.2018), and IB4 (24.10.2018). For approach see text. Number in italics are not considered
for the total volume calculation. Lateral displacement of icebergs is not considered.

| Event | Process | State | Volume above water level / 10% (m³) | Total volume / 100% (m³) |
|---|---|---|---|---|
| IBE2 | ice peninsula detachment | before detachment | 3206.8 | 32,068 |
| | ice emergence | after emergence | 2364.8 | 23,648 |
| IBE3 | ice emergence | after emergence | 3216.9 | 32,169 |
| | Ice uplift | before uplift | 7060.3 | *70,603* |
| | | after uplift | 48,369.2 | *483,692* |
| | | difference | 41,308.9 | 413,089* |
| IBE4 | ice peninsula detachment | before detachment | 2833.6 | 28,336 |
| | ice disintegration | after emergence | 50,926.8 | 509,268 |
| Sum | | | | 1,038,578 |

(* difference considered in the total)


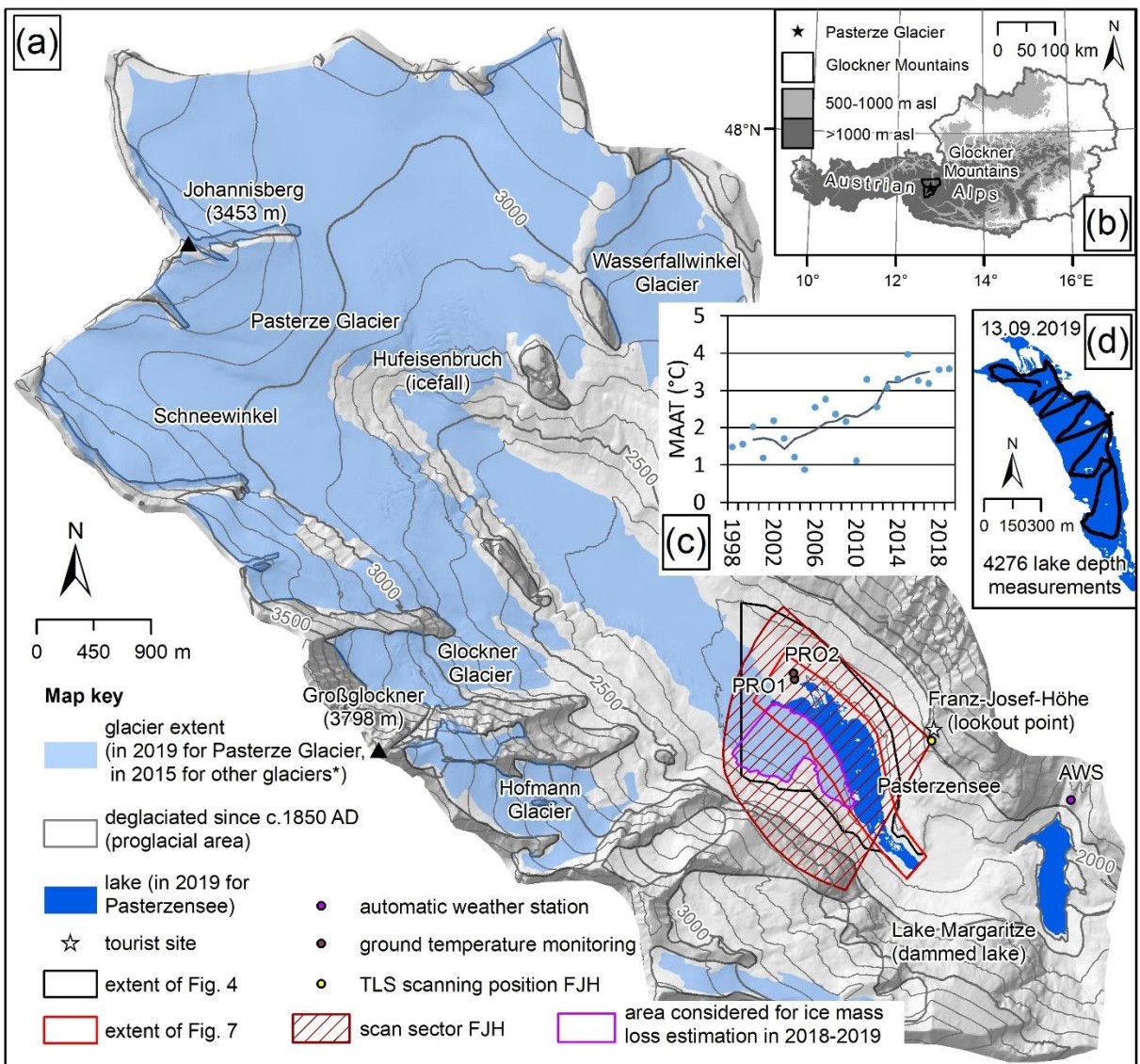


**Figure 1:** Pasterze Glacier. (a) Location of Pasterze Glacier at the foot of Großglockner (3798m asl).
Relevant sites are indicated; (b) location of the study area within Austria; (c) mean annual air
temperature (MAAT) at the automatic weather station (AWS) Margaritze in 1998-2019 (single years and
5-year running mean); (d) position of 4276 lake depth measurements carried out on 13.09.2019.
Hillshade in the background of (a) from 2012 source KAGIS. Extent of glacier and lake in 2019 this study.
Glacier extent of 2015 (*) based on Buckel and Otto (2018). Glacier extent of c.1850 based on own
mapping.

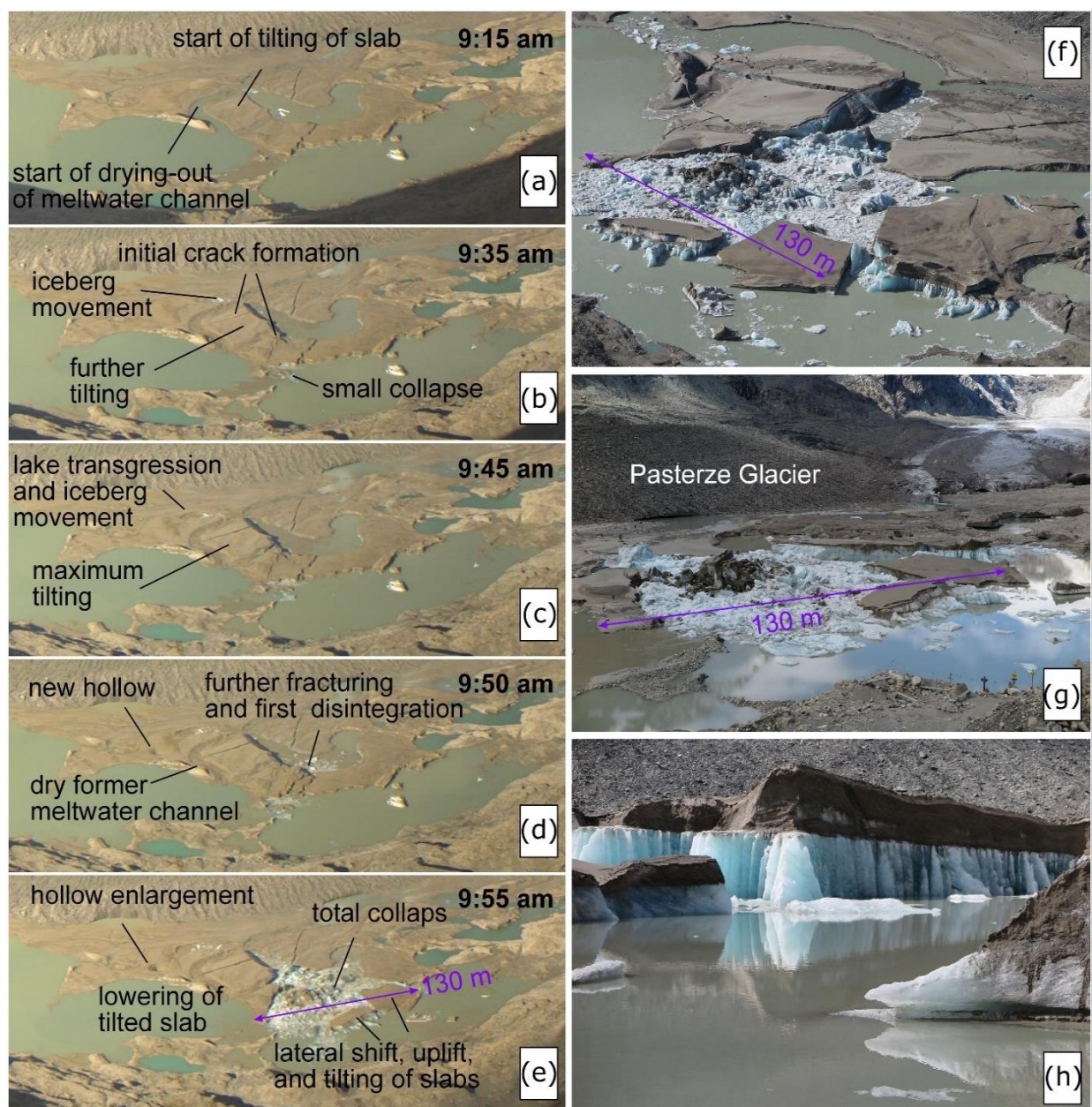


**Figure 2:** Evolution of the proglacial area at Pasterze Glacier during a period of only 40 minutes

(20.09.2016; from 9:15 to 9:55 a.m.) due to loss of hydrostatic disequilibrium and buoyancy as depicted

by an automatic time-lapse camera (a-e) and observed in the field a few hours after the event (f-h). Note

the sudden fracturing between 9:50 and 9:55 am. (a-e) provided by GROHAG, (f-h) provided by Konrad

Mariacher, 20.09.2016.



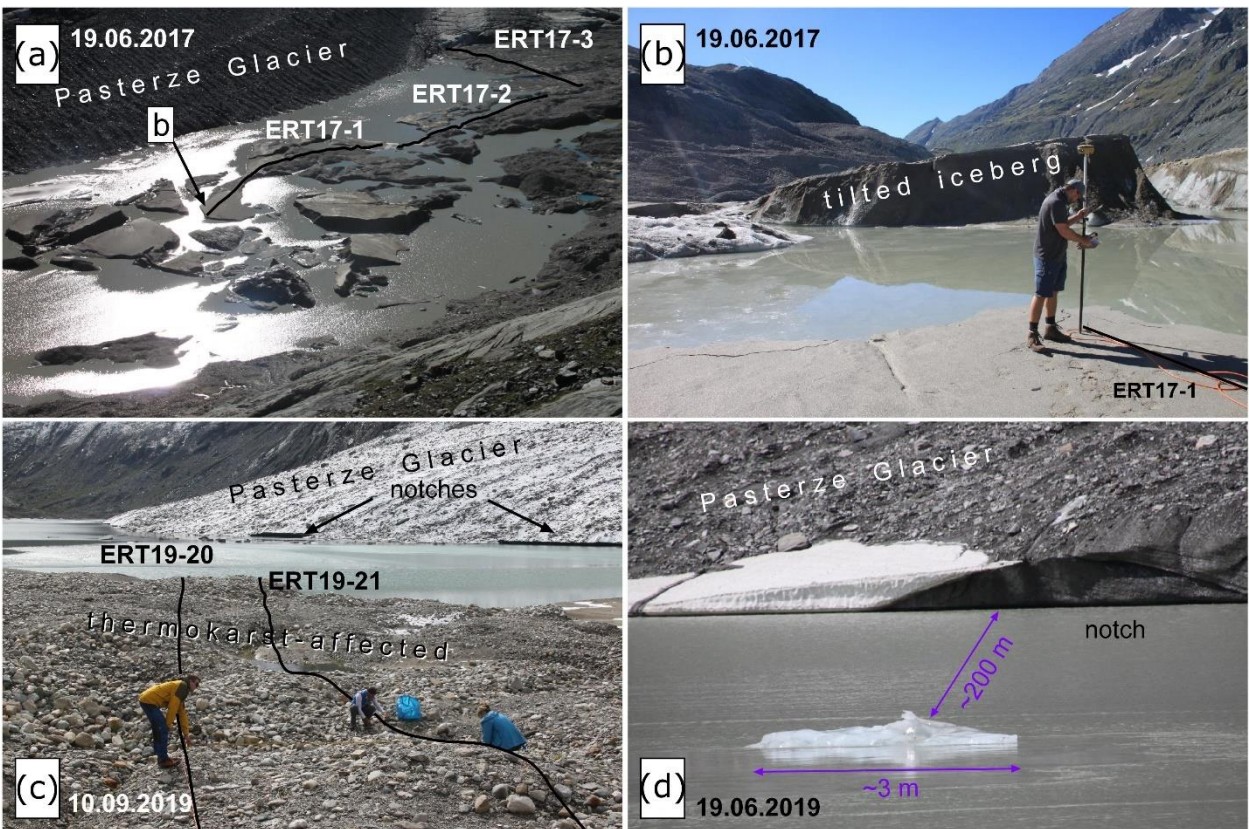

**Figure 3:** Field impressions of the ice-contact lake and its close surrounding: (a) overview image
depicting the distribution of water bodies, icebergs and debris-covered dead-ice bodies on 19.06.2017.
Courses of ERT profiles presented in Figure 9 are shown; (b) starting point of ERT17-1 surveyed by GNSS;
(c) thermokarst-affected area with courses of two ERT profiles on 10.09.2019. Note the Pasterze Glacier
and thermo-erosional notches at the lake level; (d) buoyant calving of a small iceberg ('shooter') c.200 m
from the subaerial glacier front observed during fieldwork (all photographs Andreas Kellerer-Pirklbauer).

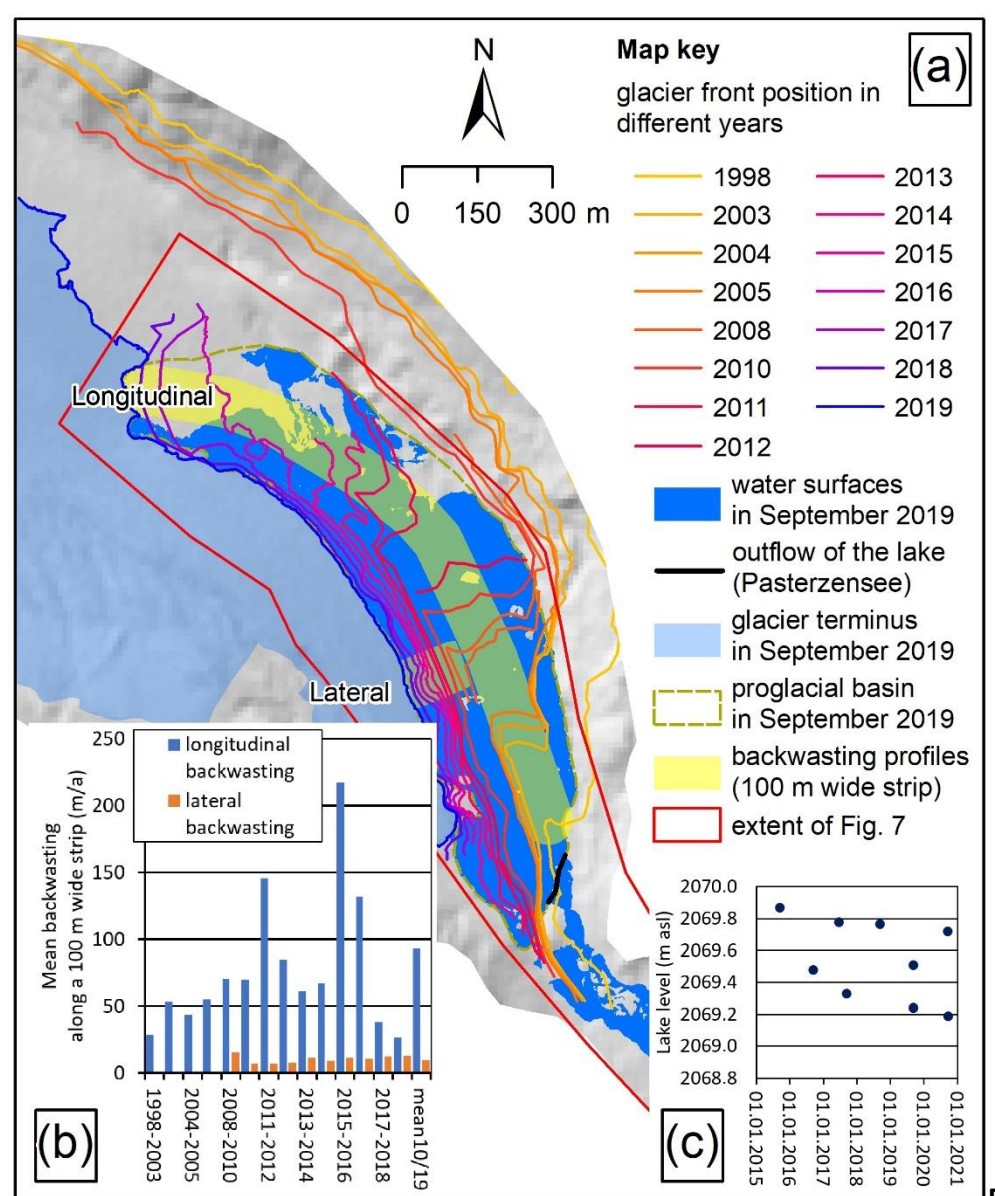


**Figure 4:** Terminus position of Pasterze Glacier for the period 1998 to 2019 and lake-level variability of Lake Pasterzensee in
the period 2015 to 2020 derived mainly from sequential GNSS data. (a) the extent of water surfaces
including the Lake Pasterzensee and the delineation of the proglacial basin is shown for September
2019. 100 m wide profiles (lateral and longitudinal) used for backwasting calculations are indicated.
Backwasting results are depicted in (b) (background hillshade based on 10m DTM, KAGIS). (c) lake level
elevations for nine stages between 17.09.2015-22.09.2020 (all between 11 am and 3 pm.

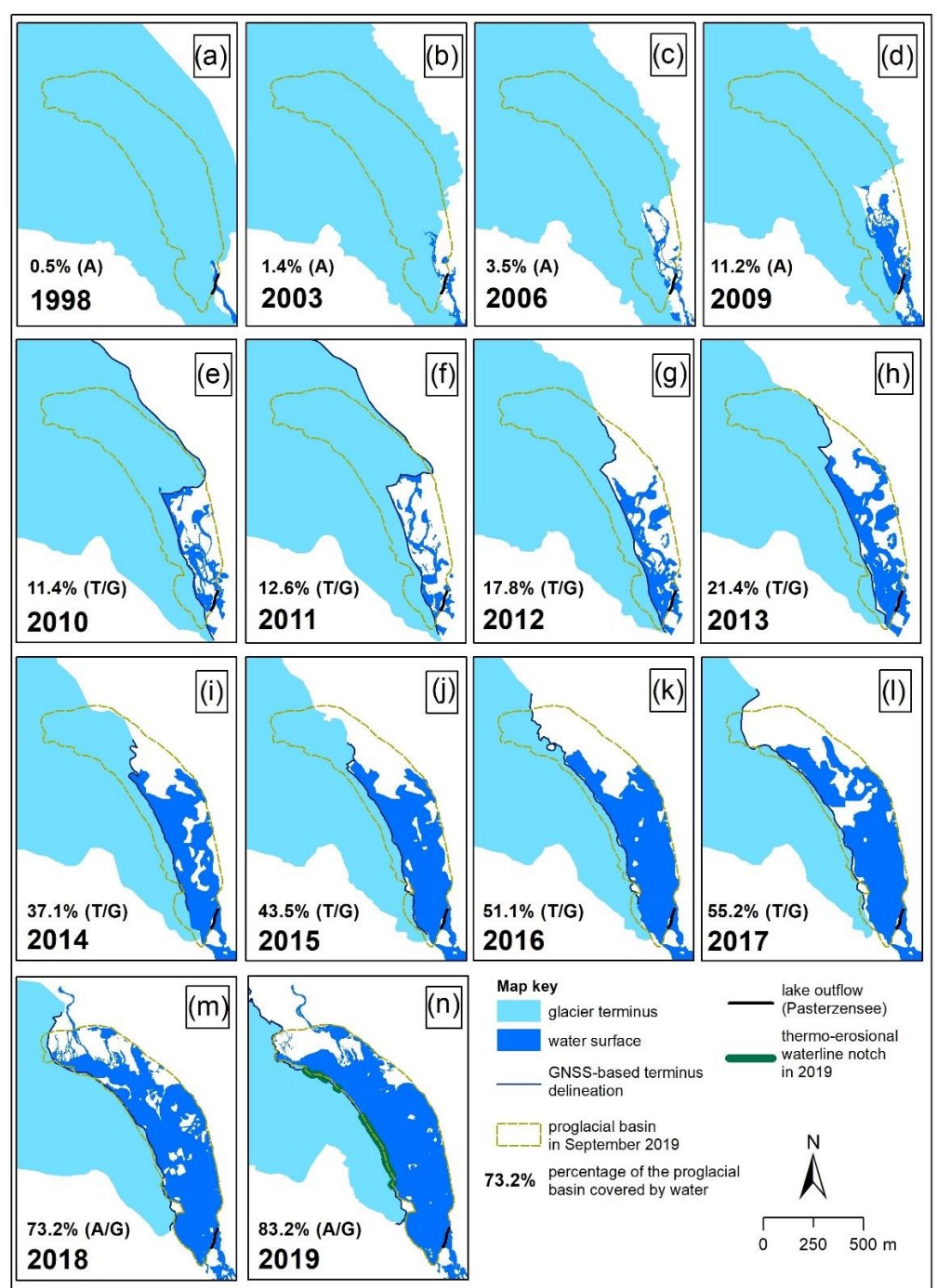

**Figure 5:** Glacier recession and evolution of proglacial water surfaces since 1998 at Pasterze Glacier. The
proglacial basin as defined for September 2019 is depicted in all maps for comparison. For data sources
refer to text and Table 1. A=airborne photogrammetry, T=terrestrial laserscanning, G=GNSS.

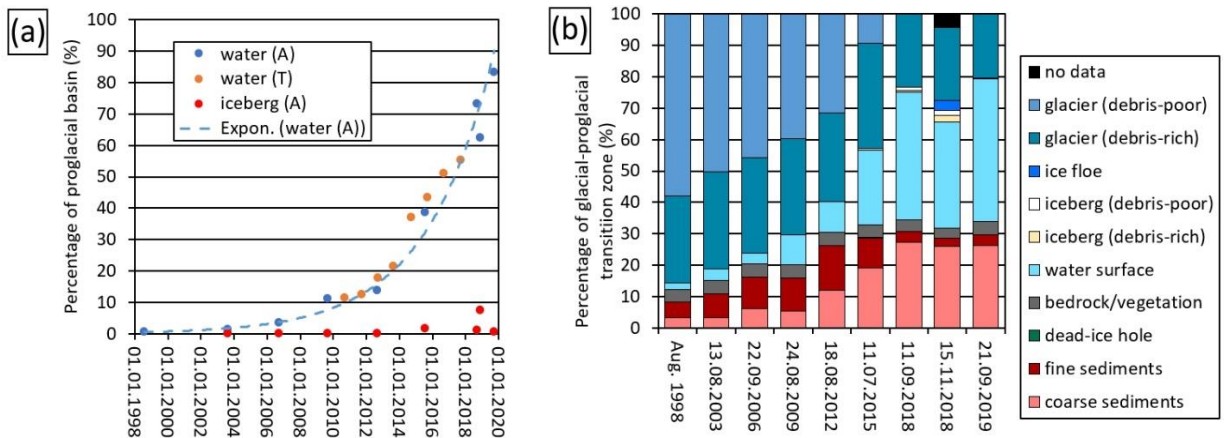


**Figure 6:** Glacial-proglacial transition zone: (a) Evolution of water surfaces and icebergs in the proglacial
basin (100%=0.37 km²; Fig. 5 for delineation) of Pasterze Glacier since 1998 based on airborne
photogrammetry/A or terrestrial laserscanning/T data. Icebergs only based on airborne
photogrammetry/A; (b) summarising graph depicting relative changes of different surface types in the
glacial-proglacial zone (100%=0.76 km²; extent as shown in Fig. 7) since 1998.




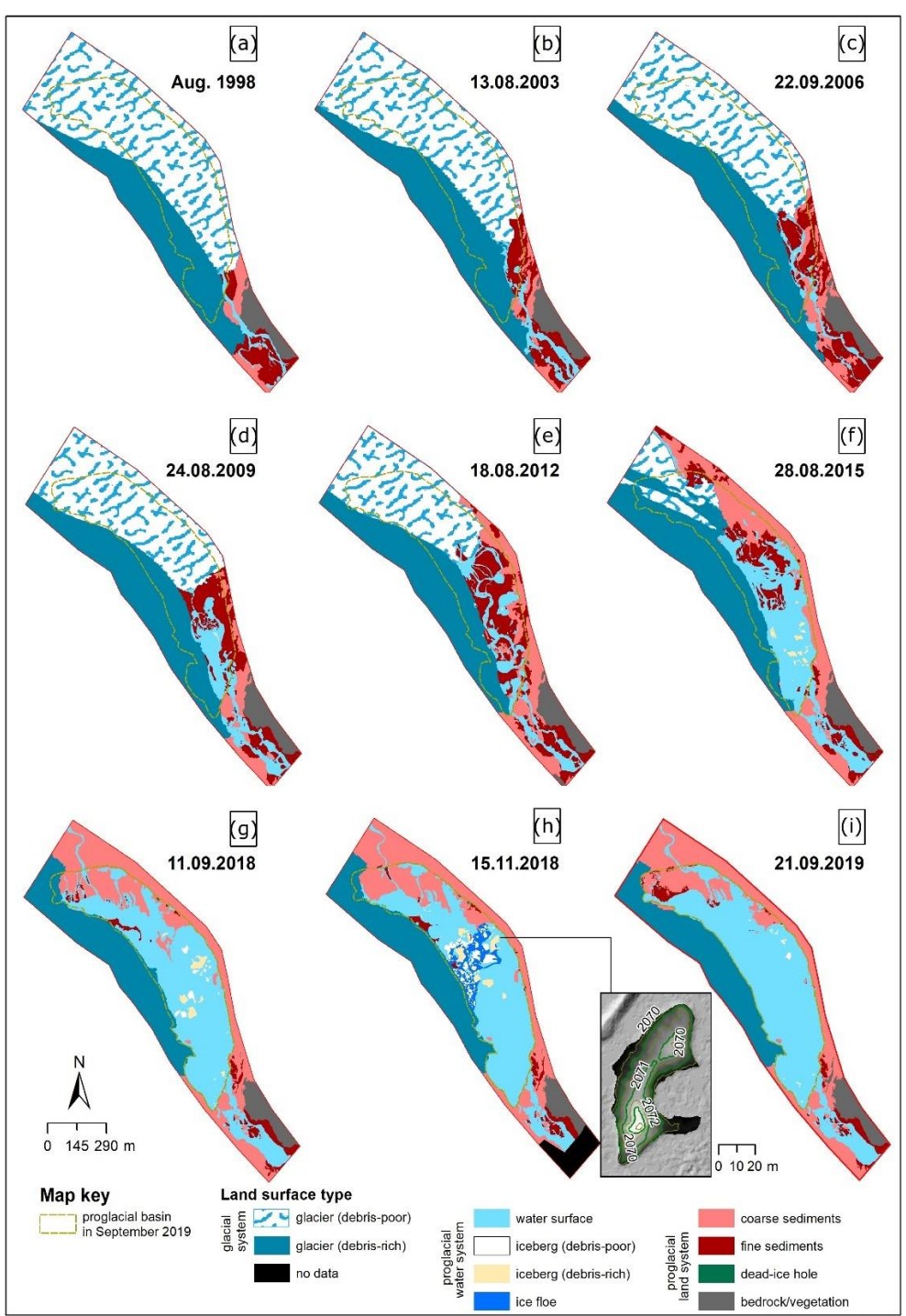


**Figure 7:** Land cover evolution in the glacial-proglacial transition zone (0.76 km²) of Pasterze Glacier

between 1998 and 2019 based on visual landform classification. The proglacial basin as defined for

September 2019 is depicted in all maps for comparison. For data sources refer to text and Table 1.

Inset map in (h) depicts a digital elevation model and contour lines (0.5 m interval) of iceberg IB1.

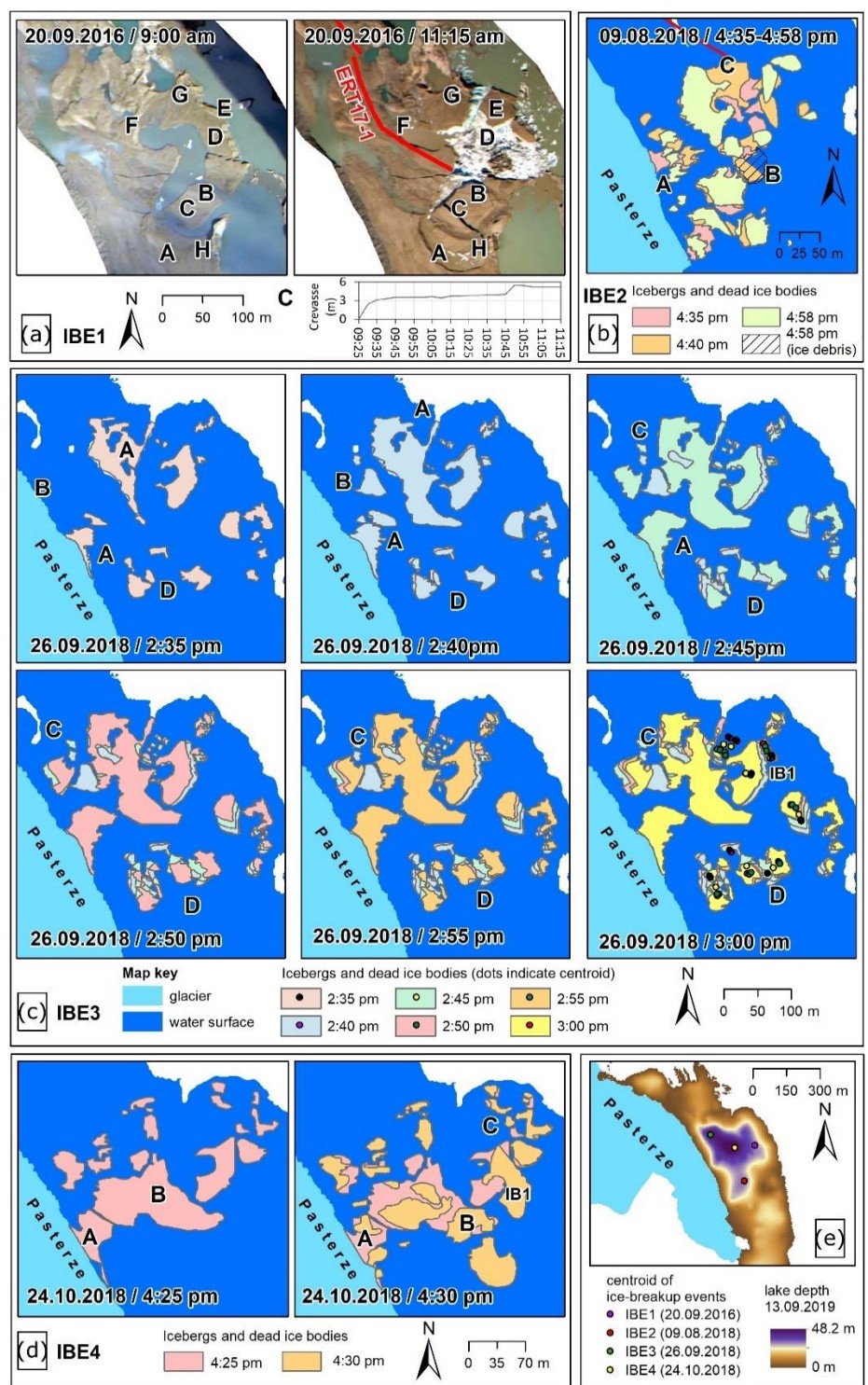


**Figure 8:** Ice-breakup events (IBE) at the ice-contact lake Pasterzensee monitored by time-lapse

photography: (a) IBE1 20.09.2016; (b) IBE2 09.08.2018; (c) IBE3 26.09.2018; (d) IBE 4 24.10.2018; (e)

overview map of the events. Capital letter in the maps indicate different processes (for details see text).


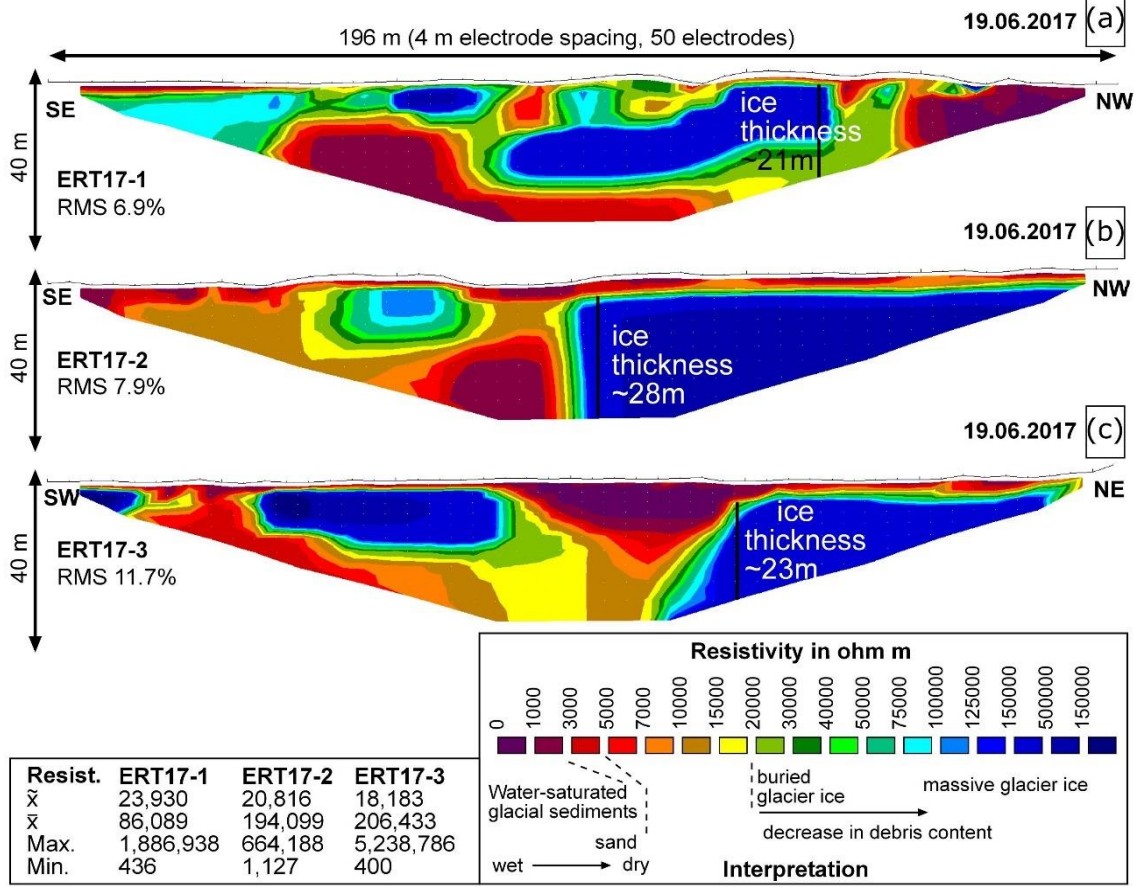


**Figure 9:** ERT results (Wenner array) and interpretation of three profiles (50 electrodes, 4 m spacing,
length 196 m) measured in the proglacial area of Pasterze Glacier on 19.06.2017 (location: Figs 3, 10).
Summary statistics in the inset table: (a) ERT17-1 – ice lens with a thickness of c.21 m; (b) ERT17-2 –ice
thickness c.28m; (c) ERT17-3 –ice thickness c.23m. For (b) and (c) - ice thickness exceeded the depth of
ERT penetration.




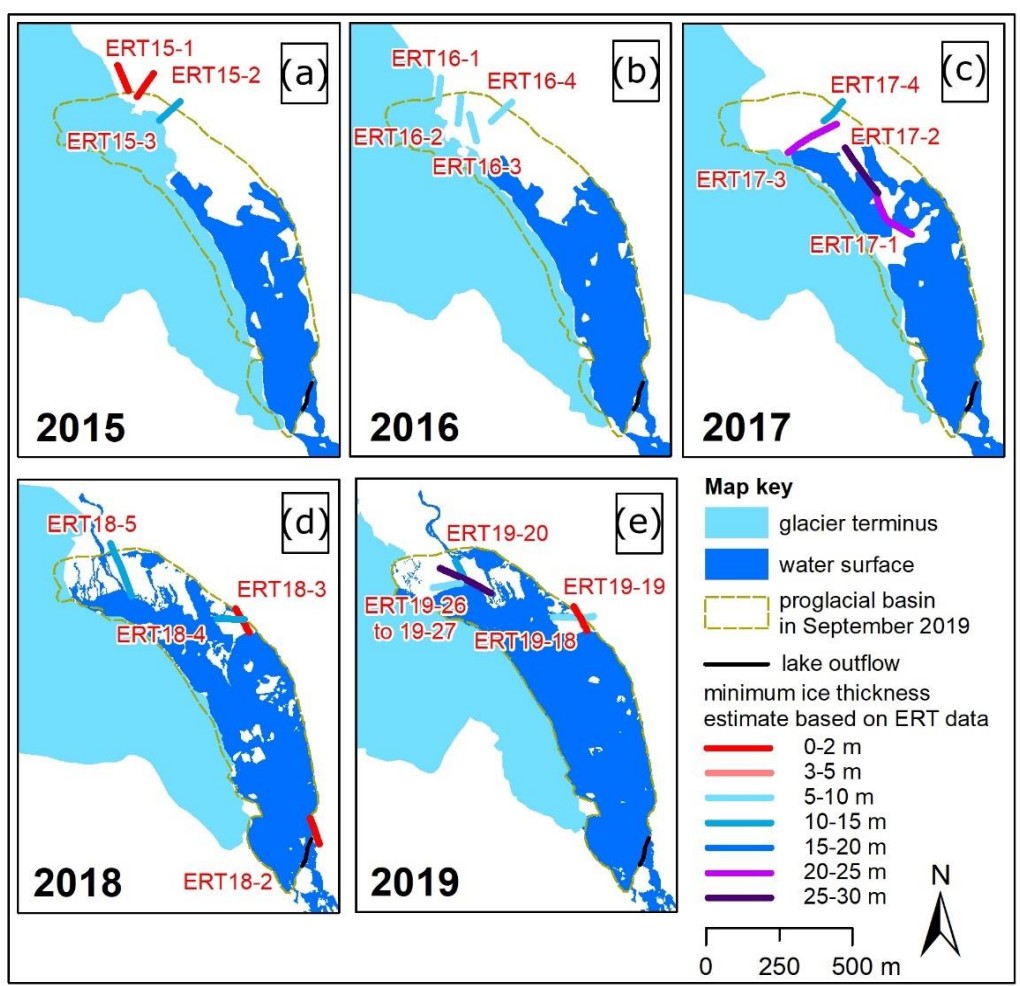

**Figure 10:** Interpreted minimum ice thicknesses based on electrical resistivity tomography (ERT) data

(for estimation approach see Fig. 9) in the proglacial area of Pasterze Glacier for (a) 30.09.2015, (b)

13.09.2016, (c) 19.06.2017, (d) 13.09.2018, and (e) 09.09.2019 as well as 10.09.2019. 'Minimum' means

in this case that the base of the ice core was commonly below the depth of ERT penetration.







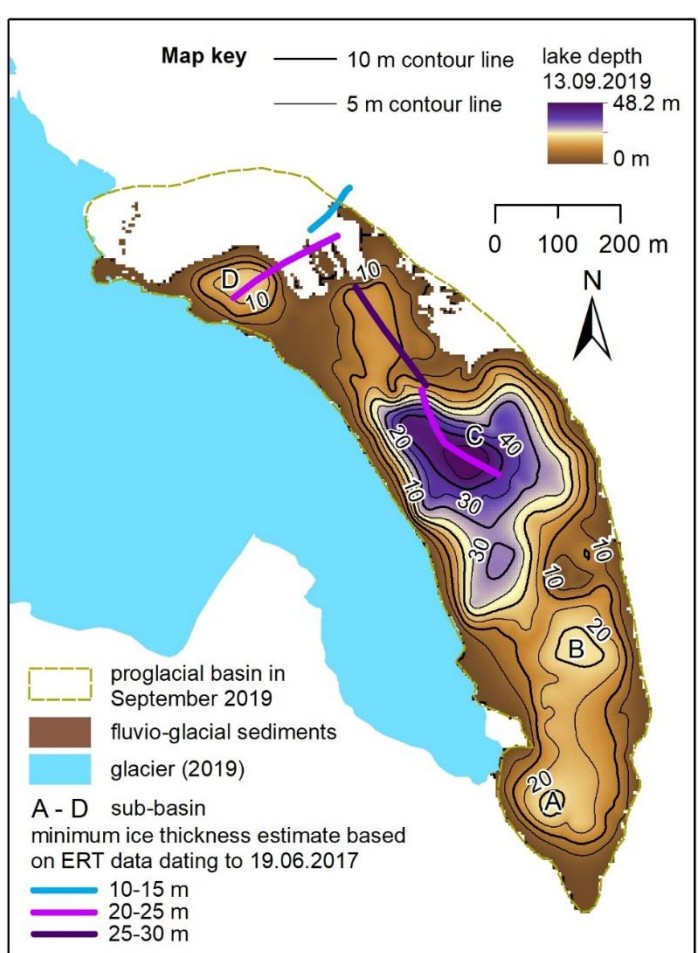


**Figure 11:** Lake bathymetry based on echo sounding data acquired in 2019 and its relationship to the
ERT data from 2017: glacier extent and lake bathymetry in September 2019 (5 m grid resolution); the
extent of the proglacial basin as defined for September 2019 is drawn in the map for orientation.


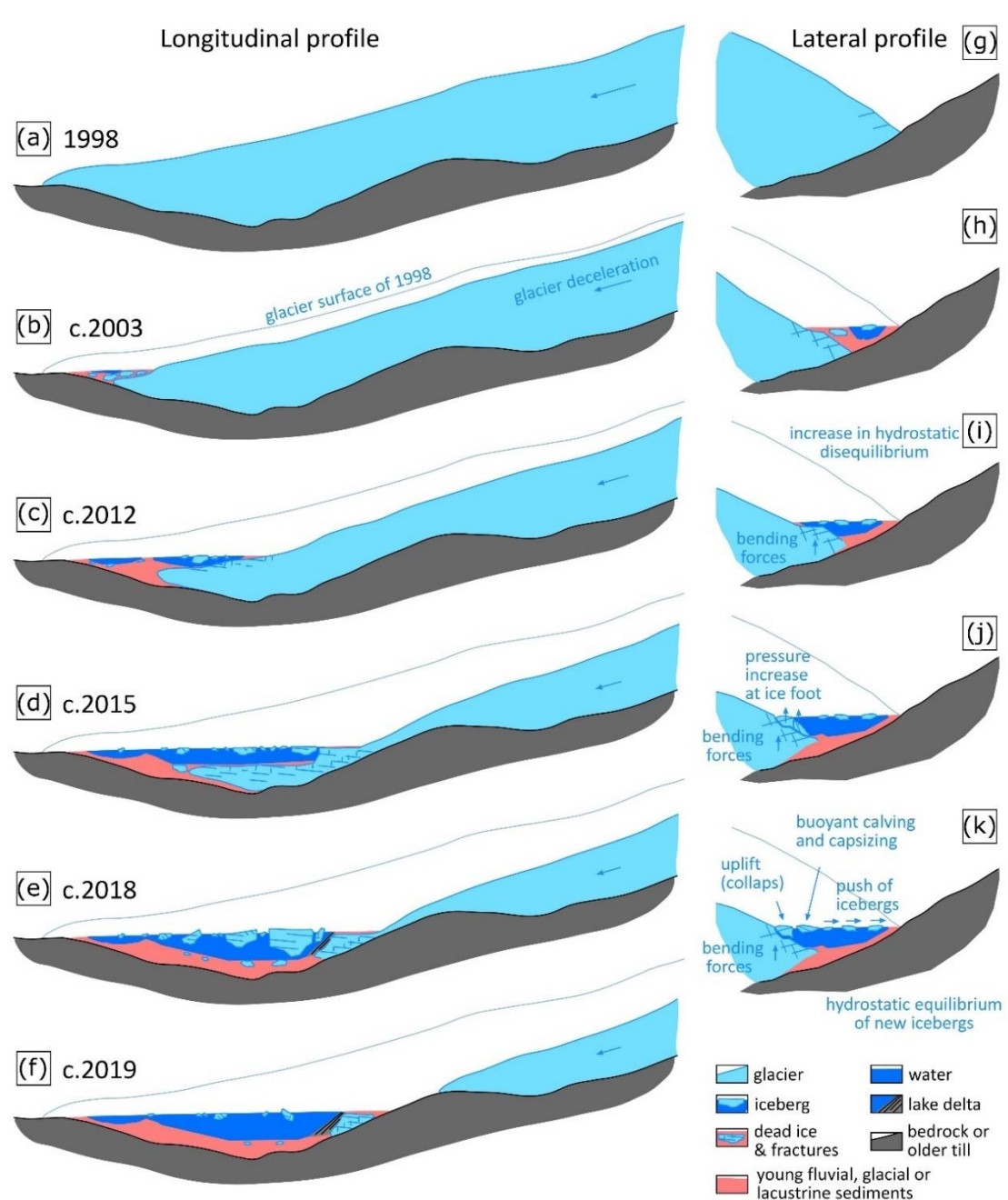


**Figure 12:** Conceptual model of the evolution of the glacial-proglacial transition zone at Pasterze Glacier

since 1998 behind a bedrock threshold: panels (a) to (f) depict changes along a longitudinal profile at the

east side (supraglacial debris-poor) of the glacier tongue; panels (g) to (k) visualize lateral changes and

related processes.




