# Peer review of "Buoyant calving and ice-contact lake evolution at Pasterze Glacier (Austria) in the period 1998-2019"

_The Cryosphere, 2020_

## Referee Comment (RC1) · Anonymous Referee #1 · 1 Dec 2020

This is a well researched observational study on the calving mechanism of a quickly retreating glacier in the European Alps. Pasterze glacier is the largest glacier in Austria, and the mechanisms described here may be pertinent to other similar alpine glaciers where strong temperature increase due to climate warming caused substantial increase in surface melting. The novel findings show new insights in a process, which to my knowledge hasn't been observed in the European Alps before. Image data from satellites and time-lapse cameras and GNSS data acquired over a period of 20 years reveal a significant glacier retreat, and the formation of supraglacial lakes and a proglacial lake; the latter increasing in size at exponential rate. The drastic retreat ('backwasting') significantly reduced once the glacier was detached from the lake (to-

wards the end of the observation period). The interpretation of the calving events relies on time lapse photography, which are put into a geophysical context by making use of geo-electrical resistivity tomography (ERT) surveys prior to the calving (2015-2019). It is concluded that debris-covered dead-ice bodies were widespread and existed in a proglacial basin (up to 48 m deep), which disintegrated in four ice-break-up events (buoyant calving). The processes related to hydrostatic disequilibrium are clearly identified, and the authors conclude that buoyant driven calving is causing the rapid formation of the pro-glacial lake. Overall, the applied methods are suitable and scientific rigour has been applied. However, I have added major comments about the referencing below. The chain of arguments is convincing and I rate the manuscript suitable for publication in The Cryosphere after addressing a small number of major and minor comments addressed below. The figures are largely of great quality, and I have only one minor comment listed below.

Major comments:

One of my main comments is about the literature and references used. Quite a number is either not quality assured, and hard or impossible to find. Some of this literature is in German, and in that case it needs in my opinion a precise reference to a page or figure with description, in order to allow a non German speaking person to follow. Examples are Avian et al. (2007), Bernsteiner (2019), Kellerer-Pirklbauer (2017), Krisch and Kellerer-Pirklbauer (2019), and Wakonigg and Lieb (1996). I couldn't find Loke (2000). These references need in my opinion get either better referenced or replaced, or otherwise, if the information is not critical, could be removed. The latter could help to shorten the manuscript, and to give more focus on the analysis and interpretation of the geophysical survey and the time-lapse photography. It is unclear if data will be provided upon publication, but it would be great to have the time-lapse photographs of ice-break-up events as animated gifs. I am unsure what 'super-buoyant' actually means, but I was wondering how the thickness of sediments influence the buoyancy of dead ice bodies. As sedimentation is identified as an important process, is it possible

to estimate if sedimentation, solely by its weight, would influence or delay the break-up event? Or is it a relatively thin layer of sediments on dead ice which results in what is described as 'super buoyant', as opposed to thick layers, which would prevent dead ice from buoyant claving? As a final comment, I agree that flood outbursts pose a significant hazard. I imagine that the sudden buoyant calving is also a significant risk for tourists walking in the pro-glacial area.

Minor comments:

Line 27: Use GNSS (preferably) or GPS throughout the document; Replace 'geoelectrical' with geo-electrical or, preferably, with 'electrical', in order to make the acronym ERT obvious. Line 35: ... for the fast lake expansion (add 'the'). Line 80-82: This sentence requires a reference, as it reports on 'observed instances of fast lake-bottom lowering'. Line 89: What does 'super-buoyant' mean? I doubt that there is something like this, and should be replaced by 'buoyant'. Line 94: Kellerer-Pirklbauer (2017) is not a quality assured reference and should be taken out or replaced, or the sentence rephrased. Line 120: Wakonigg and Lieb (1996); unaccessible source. Line 234-236: What method was used to measure lake level variation? Line 242: I could not find this thesis, and as it is in German, it is not suitable reference. Line 244: I suggest to provide a more descriptive heading; Line 249: Geotom, Geolog, Germany – provide more specific information or a suitable reference; Line 253: RTK-GNSS Line 258-259: 'Bad datum points were removed...'' What do 'bad datum' points mean in this context? Line 266: an estimated accuracy of 10 cm is likely only the case for a flat bathymetry; how is this degraded for slopes? Line 469: Pasterzesee; this is the only time that this is used in the document; change to 'Lake Pasterzensee'? Line 506: reference; Stokes et al. (2007) Line 576: what does 'super-buoyant' mean? Line 778: Roehl (2006) Line 781: Remote Sens-Basel; typo? Line 834-835 (Figure 4): one box describes 'extent of Figs 5, 7, 9'. Is this supposed to be 'Figs 5, 7, 10, 11'? Or, more likely, is it only referring to Fig. 7?

---

## Referee Comment (RC2) · Anonymous Referee #2 · 1 Dec 2020

General comments

The paper 'Buoyant calving and ice-contact lake evolution at Pasterze Glacier (Austria) in the period 1998-2019' by Kellerer-Pirklbauer et al. presents important insights in new type of processes appearing during the present phase of rapid glacier recession in the Alps. The multimethod and long term investigation of the formation of lakes with ice contact, relocation of debris and calving events is key for estimating present and future retreat rates not only in the Alps, but in all mountain regions where the over-deepened glacier tongues disintegrate. The overall presentation is well structured and clear, the language is quite free of spelling and grammar errors and clear. What actually

is missing and would be very helpful, is the quantification of loss by calving during the period to the total ablation at the glacier tongue, showing how large the contribution of this new process actually is. This would be nice to read in the abstract also, just for example the specific mass loss/year at the lake and the mean direct specific surface mass balance at areas in the same elevation without contact to the lake.

Specific comments

145: are you referring to a calendar year or a mass balance year? What exactly would be the implication of the temperatures during the winter?

Technical corrections

233: pixels? 235: 0.95 m 236: .Thus, …? 266: 0.1 m? 283: of about 1.4 km 362: MEZ?, pm missing at the end of the line 441: 4 106 m3? Figure 4: please check again the legend, you use a thin black line outlining the hillshade, and at the same time for the outflow

---

## Author Comment (AC1) · 8 Jan 2021

Comments on the manuscript with the id "tc-2020-227" Entitled "Buoyant calving and ice-contact lake evolution at Pasterze Glacier (Austria) in the period 1998–2019" by Andreas Kellerer-Pirklbauer, Michael Avian, Douglas I. Benn, Felix Bernsteiner, Philipp Krisch, and Christian Ziesler

Reviewer # 1

[1] General comments Reviewer: This is a well researched observational study on the calving mechanism of a quickly retreating glacier in the European Alps. Pasterze

glacier is the largest glacier in Austria, and the mechanisms described here may be pertinent to other similar alpine glaciers where strong temperature increase due to climate warming caused substantial increase in surface melting. The novel findings show new insights in a process, which to my knowledge hasn't been observed in the European Alps before. Image data from satellites and time-lapse cameras and GNSS data acquired over a period of 20 years reveal a significant glacier retreat, and the formation of supraglacial lakes and a proglacial lake; the latter increasing in size at exponential rate. The drastic retreat ('backwasting') significantly reduced once the glacier was detached from the lake (towards the end of the observation period). The interpretation of the calving events relies on time lapse photography, which are put into a geophysical context by making use of geo-electrical resistivity tomography (ERT) surveys prior to the calving (2015-2019). It is concluded that debris-covered dead-ice bodies were widespread and existed in a proglacial basin (up to 48 m deep), which disintegrated in four ice-break-up events (buoyant calving). The processes related to hydrostatic disequilibrium are clearly identified, and the authors conclude that buoyant driven calving is causing the rapid formation of the pro-glacial lake. Overall, the applied methods are suitable and scientific rigour has been applied. However, I have added major comments about the referencing below. The chain of arguments is convincing and I rate the manuscript suitable for publication in The Cryosphere after addressing a small number of major and minor comments addressed below. The figures are largely of great quality, and I have only one minor comment listed below. Reply by authors: Thank you very much for these general comments!

[2] Detailed comments - Major comments: Reviewer: One of my main comments is about the literature and references used. Quite a number is either not quality assured, and hard or impossible to find. Some of this literature is in German, and in that case it needs in my opinion a precise reference to a page or figure with description, in order to allow a non German speaking person to follow. Examples are Avian et al. (2007), Bernsteiner (2019), Kellerer-Pirklbauer (2017), Krisch and Kellerer-Pirklbauer (2019), and Wakonigg and Lieb (1996). Reply by authors: The reference list was revised and

only the most important German-written references were kept. Furthermore, precise page/figure referencing was added in German-written articles where it was necessary and appropriate.

Reviewer: I couldn't find Loke (2000). These references need in my opinion get either better referenced or replaced, or otherwise, if the information is not critical, could be removed. The latter could help to shorten the manuscript, and to give more focus on the analysis and of the geophysical survey and the time-lapse photography. Reply by authors: The methodological description of the ERT-method was modified to "The apparent resistivity data were inverted in Res2Dinv using the robust inversion modelling. ERT data were checked before processing for abnormally high or low resistivity values. Abnormal values are commonly related to measurement errors and/or bad electrode contact usually visible at all depths. Such 'bad datum points' were excluded manually (Kneisel and Hauck, 2008). The number of iterations was stopped when the change in the RMS error between two iterations was small."

Reviewer: It is unclear if data will be provided upon publication, but it would be great to have the time-lapse photographs of ice-break-up events as animated gifs. Reply by authors: Some of the major data sets used in this study will be submitted to a reliable repository which uses the DOI system. In doing so, we thought about the well-known repository Pangaea where we published already earlier data sets. In detail, data sets presumably accepted by Pangaea should be: (i) terminus position of Pasterze Glacier for the period 1998 to 2019 derived primarily from sequential GNSS data; (ii) extent of proglacial water surfaces between 1998 and 2019; and (iii) lake depth data based on echo sounding acquired on 13.09.2019. Furthermore, we intend to publish in the supplementary material two animations related to the time lapse photographs. The permission to use time-lapse images for such animations have been kindly granted by the GROHAG company. The first animations will show the general evolution of the proglacial lake between 2010 and 2020. The second animation will show in detail the ice-breakup event which occurred on the 20.09.2016.

Reviewer: I am unsure what 'super-buoyant' actually means, but I was wondering how the thickness of sediments influence the buoyancy of dead ice bodies. Reply by authors: To make this term clearer, we changed the relevant text passage from "Ablation of lake-terminating glaciers may lead to the development of submerged ice feet or thinning of ice margins below the point of hydrostatic equilibrium. Rises in lake level can have similar results. In such cases, ice becomes super-buoyant and subject to net upward buoyant forces, promoting fracture propagation and calving (Benn et al., 2007). Calving by this process has been described by Holdsworth (1973), Warren et al. (2001) and Boyce et al. (2007)."

to

"Buoyant calving occurs where ice is subject to net upward buoyant forces sufficient to overcome its tensile strength. Such forces can develop where either ice thinning (e.g. via surface ablation) or water deepening (e.g. rises in lake level) cause the ice to become buoyant. If the ice is unable to adjust its geometry to achieve hydrostatic equilibrium it can become super-buoyant (Benn et al., 2007), creating tensile stresses at the ice base. If these stresses become sufficiently high, the ice will fracture and calve, as described by Holdsworth (1973), Warren et al. (2001) and Boyce et al. (2007). Detailed models of super-buoyancy and buoyant calving have been presented by Wagner et al. (2016) and Benn et al. (2017)."

Reviewer: As sedimentation is identified as an important process, is it possible to estimate if sedimentation, solely by its weight, would influence or delay the break-up event? Or is it a relatively thin layer of sediments on dead ice which results in what is described as 'super buoyant', as opposed to thick layers, which would prevent dead ice from buoyant claving? Reply by authors: We added in the discussion section of the manuscript: "Our field observations show that sediment is present on top of dead ice, presumably thicker at the north-western end of the lake where the main glacial stream enters the lake. Sediment cover will affect the buoyant weight of the ice column, potentially offsetting buoyant forces and inhibiting calving. It is not possible to quantify

this effect, due to limited data on sediment and ice thicknesses. It is clear, however, that although sediment cover will have delayed the onset of buoyant calving, it was insufficient to prevent it in this case."

Reviewer: As a final comment, I agree that flood outbursts pose a significant hazard. I imagine that the sudden buoyant calving is also a significant risk for tourists walking in the pro-glacial area. Reply by authors: Yes, luckily, nobody was harmed so far at Lake Pasterzensee related to ice breakup events and wave formation.

[3] Detailed comments - Minor comments: Reviewer: Line 27: Use GNSS (preferably) or GPS throughout the document; Replace 'geoelectrical' with geo-electrical or, preferably, with 'electrical', in order to make the acronym ERT obvious. Reply by authors: At relevant places GPS was substituted by GNSS. This also caused modifications in Fig. 5.

Reviewer: Line 35: : : : for the fast lake expansion (add 'the'). Reply by authors: "the" was added as suggested.

Reviewer: Line 80-82: This sentence requires a reference, as it reports on 'observed instances of fast lake-bottom lowering'. Reply by authors: The value of >10 m yr-1 related to the lake-bottom lowering is based on Thompson et al (2012). This reference is added now to the sentence which is now "However, this is a slow process in which energy is conducted from the overlying water and cannot account for some observed instances of fast lake-bottom lowering with rates exceeding 10 m yr-1 (Thompson et al., 2012)."

Reviewer: Line 89: What does 'super-buoyant' mean? I doubt that there is something like this, and should be replaced by 'buoyant'. Reply by authors: See above. The clearer description of the term 'super-buoyant' is now given in the introduction section.

Reviewer: Line 94: Kellerer-Pirklbauer (2017) is not a quality assured reference and should be taken out or replaced, or the sentence rephrased. Reply by authors: In this

EGU-conference abstract the process of the sudden disintegration of debris-covered dead ice at Pasterze Glacier was first mentioned and described, therefore we consider it as important to keep this reference although it is 'only' a conference abstract.

Reviewer: Line 120: Wakonigg and Lieb (1996); unaccessible source. Reply by authors: This reference was replaced by another reference (with a DOI) where in principle the same topic is addressed.

Reviewer: Line 234-236: What method was used to measure lake level variation? Reply by authors: The text passage about lake level variations was improved by considering now a much longer period and by using direct lake level measurements acquired during our GNSS-campaigns at the glacier terminus during the last years in addition to field observations. We compared lake level data from nine different GNSS campaigns over a 5-year period (17.09.2015-22.09.2020; see Fig. 1 below). Results yield a mean value of 2069.54 m asl ranging from 2069.87 asl (17.09.2015) to 2069.19 m asl (22.09.2020) with a tendency of lake-level lowering over time. However, the minor differences suggest rather long-term stability of the lake level with elevation differences of only 68 cm primarily considering to some extent also seasonal and diurnal variations (visual example in Fig. 2). The results from the GNSS campaigns are supported by our TLS data for the period 2014 to 2019 (09.09.2014, 12.09.2015, 27.08.2016, 06.08.2017, 13.09.2018, and 03.08.2019). TLS-based lake level estimation was obtained by identifying the lowest level of the point cloud represented by the mean elevation of the lowest measurement points after plausibility check. Based on these TLS data and considering uncertainties e.g., due to error in distance measurements, we observed a lake level variation in the order of 0.8 m and a trend in lake level lowering during this period. In addition, we measured the elevation of small and fresh-looking lake terraces next to the glacier terminus on 14.09.2020 with GNSS yielding an elevation range of 59 cm. This range is also in accordance with the elevations measured by GNSS during two field campaigns on 14.09.2020 and on 22.09.2020 yielding a difference of 53 cm. Therefore, based on our long-term as well as short-term GNSS and

TLS results, we assume rather stable lake-outflow as well as lake-level conditions at least for the period 2015-2020 with a lake-level lowering trend. The assumption of lake level variations <1 m during the summer months is further supported by our field observations during the last years with the shape (stepped geometry) and size (< 1m vertical extent) of thermo-erosional notches at the waterline. Summing up, the assumption of a lake level variation in the order of <1m during the summer months seems reasonable.

Reviewer: Line 242: I could not find this thesis, and as it is in German, it is not suitable reference. Reply by authors: The mentioned reference was deleted.

Reviewer: Line 244: I suggest to provide a more descriptive heading; Reply by authors: "3.5. Geophysics" was changed to "3.5. Electrical resistivity tomography" to be more specific in the used geophysical technique. For consistency in chapter nr. 3, we believe that it is better to keep short and technical headings.

Reviewer: Line 249: Geotom, Geolog, Germany – provide more specific information or a suitable reference; Reply by authors: Additional information about the used device is now added in the text. We wrote now 'For ERT a multielectrode and multichannel system (GeoTom 2D system, Geolog, Germany) and two-dimensional data inversion (Res2Dinv) using finite difference forward modelling and quasi-Newton inversion techniques (Loke and Parker, 1996) was applied.'

Reviewer: Line 253: RTK-GNSS Reply by authors: "RTK" was modified to "RTK-GNSS"

Reviewer: Line 258-259: 'Bad datum points were removed: : :" What do 'bad datum' points mean in this context? Reply by authors: Text was modified to 'Data were checked before processing for abnormally high or low resistivity values. Abnormal values are commonly related to measurement errors and/or bad electrode contact usually visible at all depths. Such 'bad datum points' were excluded manually (Kneisel and Hauck, 2008).'

Reviewer: Line 266: an estimated accuracy of 10 cm is likely only the case for a flat bathymetry; how is this degraded for slopes? Reply by authors: The entire technical paragraph about the used sonar system was improved. We now write in the manuscript:

'Sonar measurements were carried out at Lake Pasterzensee on 13.09.2019. Water depth in the lake was measured with a Deeper Smart Sonar CHIRP+ system (depth range 0.15-100 m) consisting of an echo sounding device (single-beam echo sounder) and a GNSS positioning sensor. CHIRP stands for Compressed High Intensity Radar Pulse. We measured with 290 kHz (cone angle 16°) and a sonar scan rate of up to 15/second. According to the producer, the 16° beam angle of the 290 kHz frequency results in a ground footprint of, e.g., 0.28 m at 1 m water depth, of 2.81 m at 10 m water depth, and of 11.24 m at 40 m water depth. These footprint values are not optimal for resolving small-scale features at large water depths. However, as it was intended in this study, the footprint values are sufficient to get a first overview of the lake geometry. The accuracy of raw water-depth measurements depends on the used device, beam angle, sonar stability, bottom composition, and structure. Bandini et al. (2018) compared the Deeper Smart Sensor PROx system (precursor of CHIRP+) against the ground truth. Their results indicate a mean absolute error of 0.52 m for water depths of up to 30 m with almost perfect fit (ground truth vs. sonar) at shallow sites. The tested PROx system underestimated the water depth attributed to the beam diameter as it tends to take the shallowest point in the beam as the depth reading when going over holes or slopes. No such comparative studies are published for the CHIRP+ system. However, according to the producer the absolute error should be lower for the CHIRP+ (pers. comm. by the technical support of Deeper, 16.12.2020). In conclusion, the estimated accuracy of raw water-depth measurements should be less than 0.1 m at shallow (<5 m) and flat sites but might be as high as 0.5 m for deeper and sloping locations.

Reference to this: Bandini, F., Olesen, D., Jakobsen, J., Kittel, C. M. M., Wang, S., Garcia, M., and Bauer-Gottwein, P.: Technical note: Bathymetry observations of inland

water bodies using a tethered single-beam sonar controlled by an unmanned aerial vehicle. Hydrol. Earth Syst. Sci., 22, 4165–4181, https://doi.org/10.5194/hess-22-4165-2018, 2018.

Reviewer: Line 469: Pasterzesee; this is the only time that this is used in the document; change to 'Lake Pasterzensee'? Reply by authors: Text modified as suggested.

Reviewer: Line 506: reference; Stokes et al. (2007) Reply by authors: "Stockes" was corrected to "Stokes"

Reviewer: Line 576: what does 'super-buoyant' mean? Reply by authors: The clearer description of the term "super-buoyant" is now given in the introduction section.

Reviewer: Line 778: Roehl (2006) Reply by authors: In the mentioned paper the author uses the name "Katrin Röhl" therefore we prefer not to change the name as suggested by the reviewer.

Reviewer: Line 781: Remote Sens-Basel; typo? Reply by authors: According to the Journal Title Abbreviations by Caltech Library (https://www.library.caltech.edu/journal-title-abbreviations) this is the official journal abbreviation. REMOTE SENSING -> REMOTE SENS-BASEL

Reviewer: Line 834-835 (Figure 4): one box describes 'extent of Figs 5, 7, 9'. Is this supposed to be 'Figs 5, 7, 10, 11'? Or, more likely, is it only referring to Fig. 7? Reply by authors: Figure 4 was modified and should be clear now. The mentioned box now only refers to Fig. 7. Furthermore, Fig. 1 was also modified in accordance with these changes.
* * *
[Figure]

*Fig. 1: Lake level measurements during the period 17.09.2015-22.09.2020 (n=9) based on GNSS data acquired directly at the glacier-lake boundary. Geometric accuracy is in the range of centimeters based on comparison with nearby stable points.*

**Fig. 1.** Lake level measurements during the period 17.09.2015-22.09.2020 (n=9) based on GNSS data acquired directly at the glacier-lake boundary. Geometric accuracy is in the range of centimeters based on ...

[Figure]

*Fig. 2: Direct lake level observations in June 2017 indicating a substantial nocturnal lake level drop in the order of 50 cm during a 13.5 h period (photographs Andreas Kellerer-Pirklbauer).*

**Fig. 2.** Direct lake level observations in June 2017 indicating a substantial nocturnal lake level drop in the order of 50 cm during a 13.5 h period (photographs Andreas Kellerer-Pirklbauer).

---

## Author Comment (AC2) · 8 Jan 2021

Comments on the manuscript with the id "tc-2020-227" Entitled "Buoyant calving and ice-contact lake evolution at Pasterze Glacier (Austria) in the period 1998–2019" by Andreas Kellerer-Pirklbauer, Michael Avian, Douglas I. Benn, Felix Bernsteiner, Philipp Krisch, and Christian Ziesler

Reviewer # 2

[1] General comments Reviewer: The paper 'Buoyant calving and ice-contact lake evolution at Pasterze Glacier (Austria) in the period 1998-2019' by Kellerer-Pirklbauer et

al. presents important insights in new type of processes appearing during the present phase of rapid glacier recession in the Alps. The multimethod and long term investigation of the formation of lakes with ice contact, relocation of debris and calving events is key for estimating present and future retreat rates not only in the Alps, but in all mountain regions where the overdeepened glacier tongues disintegrate. The overall presentation is well structured and clear, the language is quite free of spelling and grammar errors and clear. Reply by authors: Thank you very much for these general comments!

[2] Major comments: Reviewer: What actually is missing and would be very helpful, is the quantification of loss by calving during the period to the total ablation at the glacier tongue, showing how large the contribution of this new process actually is. This would be nice to read in the abstract also, just for example the specific mass loss/year at the lake and the mean direct specific surface mass balance at areas in the same elevation without contact to the lake.

Reply by authors: Quantifying the ice loss by buoyant calving and comparing these losses with ablation rates at the nearby glacier tongue of Pasterze Glacier is not trivial with the available data but was attempted as explained below.

First, three of the large-scale ice-breakup events occurred between August and September 2018 (IBE2 to IBE4). For these events we tried to quantify the volume of the newly emerging icebergs as well as the volume of uplifted ice masses detaching from the subaquatic glacier ice. The latter was accomplished by comparing the calculated volume of a given ice-mass (e.g. a debris-covered ice slab) before and after the ice-breakup event. For volumetric calculations we applied the following approach. The horizontal extent of affected (newly emerged or only uplifted) ice masses was transferred back to and drawn into the original webcam images. A maximum iceberg height was also drawn as a line in the original webcam image. The length of this line was then quantified by using the ratio between the quantified horizontal extent and the marked line. The iceberg height then was obtained by applying a correction calculation for the

camera distortion produced by an incidence angle of 25° (calculated by a height difference of 310m and a horizontal distance of approx. 650m). One example for such a calculation is shown in Figure 1.

Next, the volume of individual icebergs was approximated by assuming that all ice bodies above the waterline have the form of a truncated pyramid, where A2 is 20% (for dome-shaped iceberg), 50% (for mixed iceberg type) or 80% (for tabular iceberg) of A1. The volume of truncated pyramid (iceberg above the waterline) with irregular base is given by:

$$V = h/3 * (A1 + \sqrt{(A1*A2)} + A2)$$

With A1 = area at the waterline (larger base), A2 = area of the top face (smaller base; in our cases 20, 50 or 80% of A1 depending on iceberg type), and h = maximum height of iceberg or truncated pyramid.

With this approach we quantified the volume of nine icebergs for IBE2 (09.08.2018), eight for IBE3 (26.09.2018), and two for IBE4 (24.10.2018), respectively. The volume above the waterline was then multiplied by 10 to quantify the total iceberg volume. The sums of movement affected ice masses (without lateral displacement) during the three analysed ice-breakup events was 55,717 $m^3$ for IBE2, 445,257 $m^3$ for IBE3, and 537,604 $m^3$ for IBE summing up to 1,038,578 $m^3$. We can therefore assume that ice loss by buoyant calving in the glaciological year 2018/19 (01.10.2018-31.10.2019) at Pasterze Glacier was at least in the order of 1 x 106 $m^3$. However, significant uncertainties in this quantification attempt are the visual and thus subjective estimation of the iceberg height as well as the fact that only large icebergs are considered.

Second, we quantified the ice-surface elevation changes of Pasterze Glacier where the glacier is directly attached to the proglacial lake. For this, we used two sets of TLS-Data from the 13.09.2018 and 03.08.2019. Although with this data set we do not cover the entire glaciological year 2018/19, we get an idea about direct ice mass losses at the shores of Lake Pasterze. The emergence velocity as well as the general glacier

motion at the glacier terminus is close to zero (Kellerer-Pirklbauer et al. 2008; Kellerer-Pirklbauer and Kulmer 2019) apart from ice movement related to crevasses or steeper sloping areas (Seier et al. 2017). Therefore, we can assume that surface elevation changes at the glacier terminus between the two stages equals basically glacier ablation rates. Figure 2 visualizes a quantification of surface elevation differences for a section of the lake-proximal part of Pasterze Glacier. As shown in Figure 2c, surface elevation changes and thus more or less glacier ice ablation was up to 5 m between the two stages. It was not the scope of this paper to analyze ablation rates at the terminus of Pasterze Glacier in detail. However, for a rough estimate we can calculate for the lowest part of the glacier tongue next to the proglacial lake (for extent see Figure 2d, c.0.35 $km^2$) the total ice loss for the glaciological year 2018/19. Mean ablation rates of 2.5 m or 3.0 m for this area would yield total ice losses for this area of 870,000 $m^3$ and 1,050,000 $m^3$, respectively.

To sum up, approximations of the ice volume lost by buoyant calving as well as by ablation by subaerial melting at Pasterze Glacier as shown here, seem to have been in the same order of magnitude in the glaciological year 2018/19. However, as the glaciological year 2018/19 was very unusual in terms of larger ice-breakup events (three of the four larger events occurred in this year), we can clearly conclude that glacier ice losses by buoyant calving are substantial smaller compared to subaerial ablation rates.

A condensed version of this description should be considered in the revised version of the manuscript (depending on the editor).

Mentioned references: Kellerer-Pirklbauer, A. and Kulmer, B.: The evolution of brittle and ductile structures at the surface of a partly debris-covered, rapidly thinning and slowly moving glacier in 1998–2012 (Pasterze Glacier, Austria), Earth Surf Processes, 44, 1034–1049. https://doi.org/10.1002/esp.4552, 2019.

Kellerer-Pirklbauer, A., Lieb, G. K., Avian, M., and Gspurning, J.: The response of partially debris-covered valley glaciers to climate change: The Example of the

Pasterze Glacier (Austria) in the period 1964 to 2006, Geogr Ann A, 90 A/4, 269-285, https://doi.org/10.1111/j.1468-0459.2008.00345.x, 2008.

Seier, G., Kellerer-Pirklbauer, A., Wecht, W., Hirschmann, S., Kaufmann, V., Lieb, G. K., and Sulzer, W.: UAS-based change detection of the glacial and proglacial transition zone at Pasterze Glacier, Austria, Remote Sens-Basel, 9, 549, 1-19, https://doi.org/10.3390/rs9060549, 2017.

[3] Specific comments: Reviewer: 145: are you referring to a calendar year or a mass balance year? Reply by authors: We refer here to calendar years. This is now indicated accordingly in the text.

Reviewer: What exactly would be the implication of the temperatures during the winter? Reply by authors: We are not quite sure if we understand the question as it was intended. The main idea about depicting the MAAT evolution in the study area during the rather recent past was to show general climatic changes in the study area not differentiating between summer and winter temperatures.

[4] Technical corrections Reviewer: 233: pixels? Reply by authors: Changed from "(maximum of 5 px)" to "(maximum of 5 pixels)"

Reviewer: 235: 0.95 m Reply by authors: Changed to 0.95 m,

Reviewer: 236: .Thus, : : :? Reply by authors: Modified as suggested

Reviewer: 266: 0.1 m? Reply by authors: Changed to 0.1 m as suggested.

Reviewer: 283: of about 1.4 km Reply by authors: Modified as suggested,

Reviewer: 362: MEZ?, pm missing at the end of the line Reply by authors: "pm" was added where it was missing before. We did not add the information that we speak here about the Middle European Time / MET because this addendum would then be necessary at many places in the manuscript. Furthermore, we assume that the location of the glacier makes it evident which time zone is relevant here.

Reviewer: 441: 4 106 m3? Reply by authors: $4 \times 106$ m$^3$ is correct

Reviewer: Figure 4: please check again the legend, you use a thin black line outlining the hillshade, and at the same time for the outflow Reply by authors: Figure 4 was slightly modified. The outline of the hillshade was deleted and only one black line remained.

[Figure]

*Fig. 1: Approach for iceberg height calculation exemplified for one iceberg in the ice-breakup event 3 (IBE3). Left – detail of the orthorectified webcam image with horizontal extent of iceberg, right – detail of the original webcam image with calculated horizontal extent and estimated height of iceberg. The latter data gives the ratio between the quantified horizontal extent and height. In this case the 1.5 mm in the picture correspond to 4.8 m. Finally, a correction calculation was applied accounting for the camera distortion (see text).*

**Fig. 1.** Approach for iceberg height calculation exemplified for one iceberg in the ice-breakup event 3 (IBE3). Left – detail of the orthorectified webcam ....

[Figure]

*Fig. 2: Surface elevation changes between 13.09.2018 and 03.08.2019 in a lake-proximal section of Pasterze Glacier based on TLS-data. (a) hillshade with contour lines at stage 13.09.2018, (b) hillshade with contour lines at stage 03.08.2019, (c) elevation differences between 13.09.2018 and 03.08.2019, (d) overview map of Pasterze Glacier and Lake Pasterze in 2019 also indicating the area considered in the ice-mass-loss estimation at the terminus area.*

**Fig. 2.** Surface elevation changes between 13.09.2018 and 03.08.2019 in a lake-proximal section of Pasterze Glacier based on ...